# Biodegradation of PET by the membrane-anchored PET esterase from the marine bacterium *Rhodococcus pyridinivorans* P23

Wenbin Guo [1][✉], Jingjing Duan [2][✉], Zhengguang Shi[1,3], Xue Yu[1,3] & Zongze Shao[1]

Evidence for microbial biodegradation of polyethylene terephthalate (PET) has been reported, but little is known about the PET biodegradation process and molecular mechanism by marine microorganisms. Here, we show the biodegradation of PET by the membrane-anchored PET esterase from the marine bacterium *Rhodococcus pyridinivorans* P23, elucidate the properties of this enzyme, and propose the PET biodegradation by this strain in biofilm. We identify the PET-degrading enzyme dubbed PET esterase through activity tracking. In addition to depolymerizing PET, it hydrolyzes MHET into TPA under acid conditions. We prove that it is a low and constitutively transcribed, membrane-anchored protein displayed on the cell surface. Furthermore, we also investigate the microbial groups possessing PET esterase coupled with the TPA degradation pathway, mainly in the phyla *Proteobacteria* and *Actinobacteriota*. Clarification of the microbial PET biodegradation in the marine environment will contribute to the understanding of bioremediation of marine PET pollution.

[1] Key Laboratory of Marine Biogenetic Resources, Third Institute of Oceanography, Ministry of Natural Resources, 361005 Xiamen, Fujian, China. [2] College of Environment and Ecology, Xiamen University, 361005 Xiamen, Fujian, China. [3] School of Advanced Manufacturing, Fuzhou University, 362251 Jinjiang, China. [✉]email: guo5721@tio.org.cn; duanjingjing@xmu.edu.cn

Plastics are widely produced and used in daily life due to their light weight, firmness, easy processing, low price, and good insulator. According to 'Plastic Europe' (https://www.plasticseurope.org/en), 390.7 million tons of plastics were produced worldwide in 2021, and the production of polyethylene terephthalate (PET) plastics accounted for 6.2% of plastics. Between 1950 and 2015, a total of 6.3 billion tons of primary and secondary (recycled) plastic waste was generated, of which around 9% has been recycled, and 12% incinerated, with the remaining 79% either being stored in landfills or having been released directly into the natural environment[1]. The oceanic environment has been recognized as a major sink for anthropogenic plastic waste discharged from land-based sources[2]. A particular concern is the occurrence of smaller pieces of plastic debris in the world's oceans, including those have dimensions ranging from a few μm to 500 μm (5 mm), referred to as 'microplastics'. Particles of microplastics are commonly present in seawater or sediments from coastal areas to the ocean and even in the deepest part of the world's ocean, the Mariana Trench[3–7]. It was estimated that 81.98 to 357.56 trillion particles weighed 1.11 to 4.86 million tons of floating plastics in the world's oceans in 2019[8].

In recent years, concern over plastic pollution of the marine environment and ecosystem has been raising. Marine floating microplastics will be ingested by zooplankton, which are later egested within their fecal pellets and precipitated into marine sediments as a food source for marine organisms[9]. The large plastic debris dumps accumulated in deep-sea submarine canyons are unique hot spots for environmental pollution, reflecting anthropogenic activities and biodiversity hot spots for benthic organisms[10]. Marine microorganisms can degrade long-chain plastic polymers into dissolved organic matter and particulate organic matter, which will become the food of higher-level marine plankton, thus affecting the existing state of the whole marine food web[11]. A diverse microbial community of heterotrophs, autotrophs, predators, and symbionts, referred to as the "Plastisphere", was unveiled from plastic marine debris[12].

Many plastics are transported into the marine environment each year, bringing a large amount of carbon source for marine microorganisms, which will impact their carbon assimilation pattern and affect the diversity and ecological distribution of the marine microbial population. Marine microorganisms first attach to the plastic surface to form a biofilm, then depolymerize it into smaller molecules before crossing the cell membrane[13]. In addition to the carbon cycle impacted by the assimilation of plastic, other biogeochemical cycles, including nitrogen, sulfur, iron, manganese, chromium, phosphorus, calcium, and silicate cycles, may also be impacted by the presence of plastic at sea[11]. Microplastics in coastal salt marsh sediments alter sediment microbial community composition and nitrogen cycling processes[14]. In another research, the plastisphere in estuaries exhibits a higher denitrifying activity and $N_2O$ production than surrounding bulk water, suggesting an overlooked $N_2O$ source[15].

At present, microbial degradation of plastic is a promising eco-friendly strategy, representing a great opportunity to manage waste plastics with no adverse impacts. Therefore, plastic-degrading microorganisms are considered as a tool for the bioremediation of plastic contamination in aquatic environments[16]. Until now, a large number of microbial strains have been reported being able to biodegrade PET, including bacteria, fungi, and microalgae[17]. A strain Ideonella sakaiensis 201-F6 was isolated from consortium No. 46, isolated from the samples collected at a PET-bottle recycling site in Japan, as a bacterium that was mainly involved in the degradation of PET[18]. In addition, many marine plastic degrading strains were isolated and proved to be able to grow using PET as carbon source[19–21]. PET with ester bond can be attacked by various extracellular hydrolases of microorganisms[22,23]. Cutinases from various microorganisms, including Thermobifida fusca[24], and Moniliophthora roreri[25] could carry out PET hydrolysis. In 2016, an I. sakaiensis enzyme dubbed PETase (Is-PETase) was found to efficiently depolymerize PET primarily to mono(2-hydroxyethyl) terephthalic acid (MHET) units, along with minor quantities of terephthalic acid (TPA) and bis(2-hydroxyethyl)-TPA (BHET) under mesophilic conditions[18]. From then on, scientists worldwide made great efforts on the crystal structure resolution and improving the enzyme activity and stability of Is-PETase[26–30]. Recently, a PET hydrolase from Cryptosporangium aurantiacum (CaPETase) was reported to exhibit high thermostability and remarkable PET degradation activity at ambient temperatures[31]. In addition to terrestrial habitats, PET hydrolases could also originate from marine environments[32]. In another research, a candidate PETase-like gene (SM14est) was identified in marine-sponge associated Streptomyces sp. SM14 through in silico screening[33]. A marine microbial community was isolated using Poly(butylene adipate-co-terephthalate) (PBAT) as the sole carbon source, and six putative PETase-like enzymes and four putative MHETase-like enzymes were identified[34].

Any surface in the marine environment, including plastic, will be colonized with micro- and macroorganisms[35]. The PET-colonizing microbial communities' assembly on plastics was driven by conventional marine biofilm processes and differed significantly from free-living communities[36]. Bacteria and fungi were shown to form biofilms on plastic surfaces and had some differences in biofilm formation and maturation according to the plastic type[11,37]. The 16S rRNA microbiome profiles of microbial biofilms of floating plastics and sediment-associated plastics differed significantly from sediments of the same geographic location, and the bacterial community composition was plastic nature dependent[38]. In addition to plastic type, environmental conditions, including water temperature, salinity, and pH, also play a role in the formation of a plastic-specific microbial community[39]. In a recent study, the rod-like, filamentous, and peanut-like morphologies of the corrosion structures are directly observed in the marine PE plastic surface, which are well in line with those of microorganisms, suggesting that they were derived from the biodegradation of marine microorganisms on them[40].

In this study, We identified a PET esterase from the marine bacterium Rhodococcus pyridinivorans P23 through BHET hydrolysis activity tracking and heterologous expression in E. coli BL21(DE3). We elucidated the characteristics of the PET esterase through enzyme activity determination, and revealed the PET biodegradation pathway in this strain through genome sequencing and transcriptome analysis. Further, we proposed the PET biodegradation process by R. pyridinivorans P23 in marine environments.

## Results

**Isolation and identification of a PET-degrading marine strain P23.** Several marine strains were screened from deep-sea sediment enrichment culture with PET powder as the sole carbon and energy source for growth. Among these PET-degrading strains, a gram-positive strain termed P23 (Supplementary Fig. 1) was selected for further investigation as it grows in a mineral salt medium with PET film and with TPA as the sole carbon and energy source. The strain P23 (CCTCC accession NO: CCTCC M2019609) showed the highest 16S rRNA gene sequence similarity to Rhodococcus pyridinivorans DSM44555[(T)] (99.93%) in EzbioCloud database. Average Nucleotide Identity (ANI) value of 95.16% was obtained via ANI Calculator in EzbioCloud using strain P23 and R. pyridinivorans DSM44555[(T)]

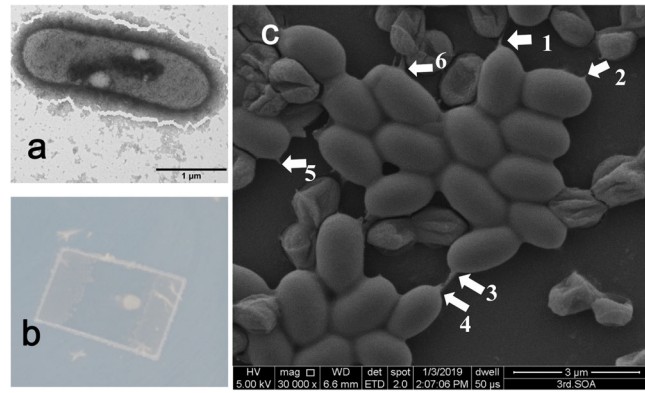

**Fig. 1 *R. pyridinivorans* P23 and its growth on PET film. a** TEM image of *R. pyridinivorans* P23 cell grown on 2216 marine agar medium. **b** The growth of *R. pyridinivorans* P23 on mineral salt agar medium covered with PET film (Good Fellow GF25214475, thickness 0.25 mm, size 20 × 15 mm) forming biofilm at 30 °C. **c** SEM images of *R. pyridinivorans* P23 cells grown on PET film for 7 days. Scale bars, 3 μm. Arrowheads indicate contact points of cell appendages and the PET film surface.

(GCA_900105195.1) genomes. A phylogenetic tree showed that strain P23 was more related to type strain *Rhodococcus pyridinivorans* DSM44555[(T)] than other type strains of *Rhodococcus* species (Supplementary Fig. 2). Therefore, strain P23 was designated as *Rhodococcus pyridinivorans*. Cells of *Rhodococcus pyridinivorans* P23 are Gram stain-positive, non-motile, and rod-shaped (Fig. 1a). During the cultivation of *R. pyridinivorans* P23 on a mineral salt medium agar plate with PET film covered on the agar, a visible biofilm on the surface especially the edge of the PET film was formed (Fig. 1b). When the biofilm was observed under a scanning electron microscope (SEM), some appendages of the cells were observed between the cells and the PET film; these might assist in the attachment of cells to the PET film (Fig. 1c).

**Biodegradation of PET film and determination of biodegradation products released by *R. pyridinivorans* P23.** When *R. pyridinivorans* P23 was cultivated in mineral salt medium with a piece of PET film (8.0 cm × 8.0 cm × 0.013 mm) as sole carbon and energy source in a flask for 7 days, an increase in the strain biomass concentration, i.e., $OD_{600}$ from 0.08 after inoculation to 0.34 at the end of cultivation was observed (Fig. 2a). After 5 weeks of cultivation of *R. pyridinivorans* P23, totally 4.03 mg of PET weight loss accounted for 4.28% was achieved (Fig. 2b). High performance liquid chromatography (HPLC) was applied to investigate the products released from the PET film in the flask cultivation, spectrum showed that TPA (retention time 10.2 min) and MHET (11.2 min) were the main released products in all the three samples, 8 h, 30 h and 7 d, while BHET (12.0 min) was only detected in 8 h with less content as it could be easily transformed into MHET by *R. pyridinivorans* P23 cells (Fig. 2c). Only tiny amount of MHET and TPA could be detected in the 7 d cultivation sample of negative control without inoculation of *R. pyridinivorans* P23 (Fig. 2c).

**Tracking of *R. pyridinivorans* P23 PET esterase activity.** In the cultivation of *R. pyridinivorans* P23 with BHET (0.2 mg/ml) or MHET (0.5 mg/ml) as the sole carbon and energy source in pH 7.0 for 120 h, the strain biomass concentration, i.e., $OD_{600}$ were both slowly increased (Supplementary Fig. 3a, b). In the BHET cultivation, BHET was degraded into MHET rapidly as soon as the inoculation of *R. pyridinivorans* P23 cells, which was prepared from 2216 marine broth cultivation in a test tube (Supplementary

Fig. 3a). Obviously, the BHET hydrolase was very likely to be present in the inoculum. In the MHET cultivation, a gradual degradation of MHET was observed accompanying the growth of this strain (Supplementary Fig. 3b). Due to the large molecular weight of PET, it is difficult to transport through the cell membrane. The PET hydrolase must be either secreted outside the cells or displayed on the cell membrane facing outside. Commonly, PET hydrolase also possesses BHET hydrolase activity due to the similarity of structures of the two substrates. In order to determine the exact location of PET hydrolase, *R. pyridinivorans* P23 was first cultivated in 2216 marine broth medium for 28 h (Supplementary Fig. 3c), sampled from 16 to 28 h with 4 h interval, centrifuged to separate the cells and supernatant and then subjected to BHET hydrolase activity determination. The MHET accumulation increased from 0.31 mM to 0.50 mM in 16 to 28 h cultured cells due to the increased number of cells sampled, and no MHET production was detected in the supernatant sample (Supplementary Fig. 4). Obviously, the BHET hydrolase (PET hydrolase) on the cell membrane of *R. pyridinivorans* P23 exhibited definite BHET hydrolase activity, while the supernatant did not.

**Genome sequencing of *R. pyridinivorans* P23 and PET hydrolase analysis.** To explore the PET hydrolase and the biodegradation pathway of the PET, the complete genome of strain *R. pyridinivorans* P23 was constructed based on co-assembly of short reads (150 bp × 2) and long reads (average 402,955 bp) derived from Illumina Hiseq and PacBio sequencing, respectively. The complete genome of *R. pyridinivorans* P23 consists of a circle chromosome of 5,295,424 bp with 68.20% GC content and four plasmids of 580,882 bp (plasmid A, 64.00% GC content), 249,161 bp (plasmid B, 63.37% GC content), 177,490 bp (plasmid C, 65.96% GC content), and 65,561 bp (plasmid D, 64.78% GC content), respectively (Supplementary Fig. 5). The genome of *R. pyridinivorans* P23 encodes 5,647 proteins, 12 rRNAs, and 55 tRNAs. The complete genome sequence of *R. pyridinivorans* P23 has been submitted to GenBank with the accession numbers CP113798-CP113802. Totally 1,044 proteins with 0–24 transmembrane domains were found through the TMHMM server V2.0 (http://www.cbs.dtu.dk/services/TMHMM/). As PET is a synthetic polymer consisting of ester bond-linked terephthalate and ethylene glycol, esterases, lipases, cutinases, and alpha/beta-hydrolases are thought to hydrolyze PET through the ester bond. In these transmembrane proteins, eight proteins were found to be designated as esterase (OQN32_06240), cutinase (OQN32_01115), or alpha/beta-hydrolase (OQN32_03920, OQN32_04705, OQN32_07550, OQN32_07555, OQN32_15860, OQN32_18580) displaying outside the cell membrane (Supplementary Table 1, Supplementary Fig. 6).

**Heterologous expression, function identification, and sequence analysis of PET esterase OQN32_06240.** After removing the transmembrane domains, the eight gene candidates of PET hydrolase were commercially synthesized with codon optimization and cloned into pET-30a vector for expression in *Escherichia coli* BL21(DE3) (Sangon, Shanghai, China). Cells of *E. coli* BL21 (DE3) mutant strains harboring these genes were induced by 0.5 mM IPTG at 20 °C, collected by centrifugation, and disrupted by sonication. The cell lysates were then subjected to SDS-PAGE analysis and BHET hydrolysis for activity confirmation. Results showed that the eight proteins were all successfully synthesized in soluble form in the supernatant of *E. coli* lysates, and only OQN32_06240, an esterase, could hydrolyze BHET (Supplementary Fig. 7). After that, heterologous expression of PET esterase (OQN32_06240) with 6×His tag was purified by Ni-affinity

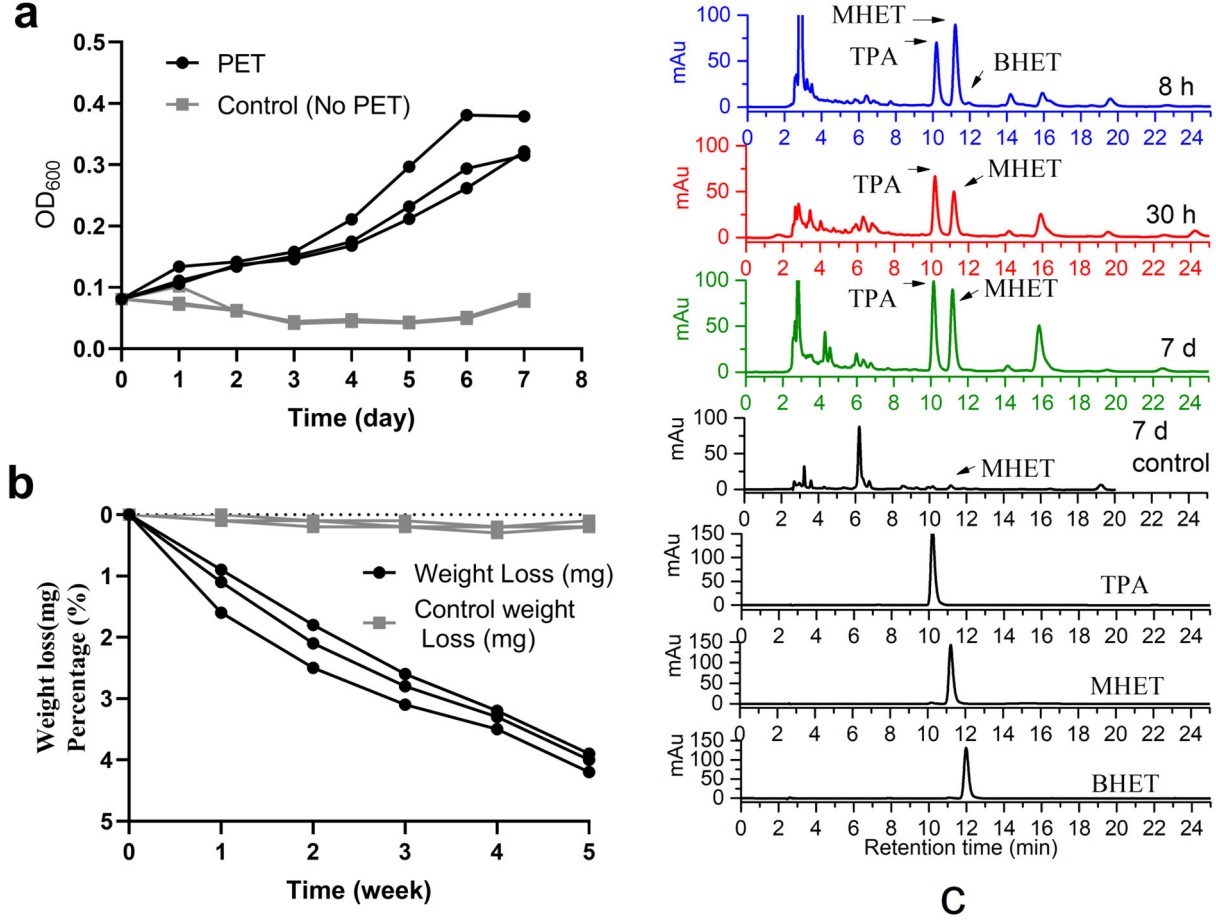

**Fig. 2 Degradation of PET by *R. pyridinivorans* P23. a** Time course of cell growth in mineral salt medium with or without PET film (Good Fellow GF89357619, thickness 0.013 mm) as sole carbon and energy source in flask shaking for 7 days. $n = 3$ biologically independent experiments were performed. Black circle for PET groups and gray square for the control groups without PET. **b** Time course of PET film degradation by *R. pyridinivorans* P23 in 5 weeks cultivation. $n = 3$ biologically independent experiments were performed. Black circle for PET groups and gray square for the control groups without inoculation. **c** HPLC spectra of culture supernatants from flask cultivation of *R. pyridinivorans* P23 grown in mineral salt medium with PET film for 8 h, 30 h, and 7 d, respectively. Flask cultivation of PET film without inoculation of *R. pyridinivorans* P23 for 7 days was set as negative control. The sample names were indicated along with the HPLC spectra.

chromatography technique. SDS-PAGE electropherogram showed that it was a homogeneous protein with molecular weight of 34 kDa (Fig. 3a). Predicted from TMHMM 2.0 server, PET esterase (OQN32_06240) harbored a 1–12 amino acid residues domain inside, a 13–35 amino acid residues transmembrane domain and a 36–340 amino acid residues catalytic domain displayed outside the cell membrane (Supplementary Fig. 8). The three-dimensional structure of PET esterase (OQN32_06240) was predicted with AlphaFold2 and found to be most identical to esterase 2 (EST2) from *Alicyclobacillus acidocaldarius*[41] (Supplementary Fig. 9). The amino acid sequences of PET esterase (OQN32_06240) and esterase 2 (EST2, 1EVQ) were aligned with Mega 6.0. Based on the study of this thermophilic Esterase[41], the amino acid residues ($S_{155}D_{252}H_{282}$ in 1EVQ and $S_{175}D_{277}H_{313}$ in OQN32_06240) forming a catalytic triad (shaded in yellow) were identified (Supplementary Fig. 8). The GXSXG motif conserved in α/β-fold hydrolases such as esterases and lipases was overlined (Supplementary Fig. 8). Multiple amino acid sequences of PET esterase (OQN32_06240) and known PET hydrolytic enzymes were aligned with Mega 6.0, and the guide tree was obtained based on the neighbor-joining method. From the phylogenetic tree, PET esterase (OQN32_06240) belonging to the Abhydrolase_3 subfamily was

found to be distinct from MHETase (tannase family), and known PET hydrolytic enzymes, including *Is*-PETase (Abhydrolase_5) (Fig. 3b).

**Characterization of PET esterase (OQN32_06240) towards PET film.** We tested the enzymatic activity of PET esterase (OQN32_06240) with substrates BHET and PET film, respectively. When PET film was used as the substrate of PET esterase (OQN32_06240) at 30 °C, the main products were TPA, MHET and BHET in pH 3.0–5.0, while only MHET and BHET were detected in pH 5.5–8.0 (Fig. 4a). PET esterase (OQN32_06240) showed relatively high enzymatic activity in pH 4.0–5.0 and best at pH 4.5 with totally 0.92 μM products (0.37 μM TPA, 0.48 μM MHET and 0.07 μM BHET) released (Fig. 4a, e). The characteristic of PET esterase (OQN32_06240) with good activity under acidic conditions differs from *Is*-PETase, which is more active under alkaline conditions[18]. As PET esterase (OQN32_06240) is a transmembrane protein displaying on the cell surface of *R. pyridinivorans* P23, the strain could be used as whole-cell biocatalyst towards PET film and BHET. When PET film was used as substrate at 30 °C with whole-cell biocatalyst at $OD_{600}$ of 1.0, the enzymatic activity in pH 3.0–4.5 was relatively high with more

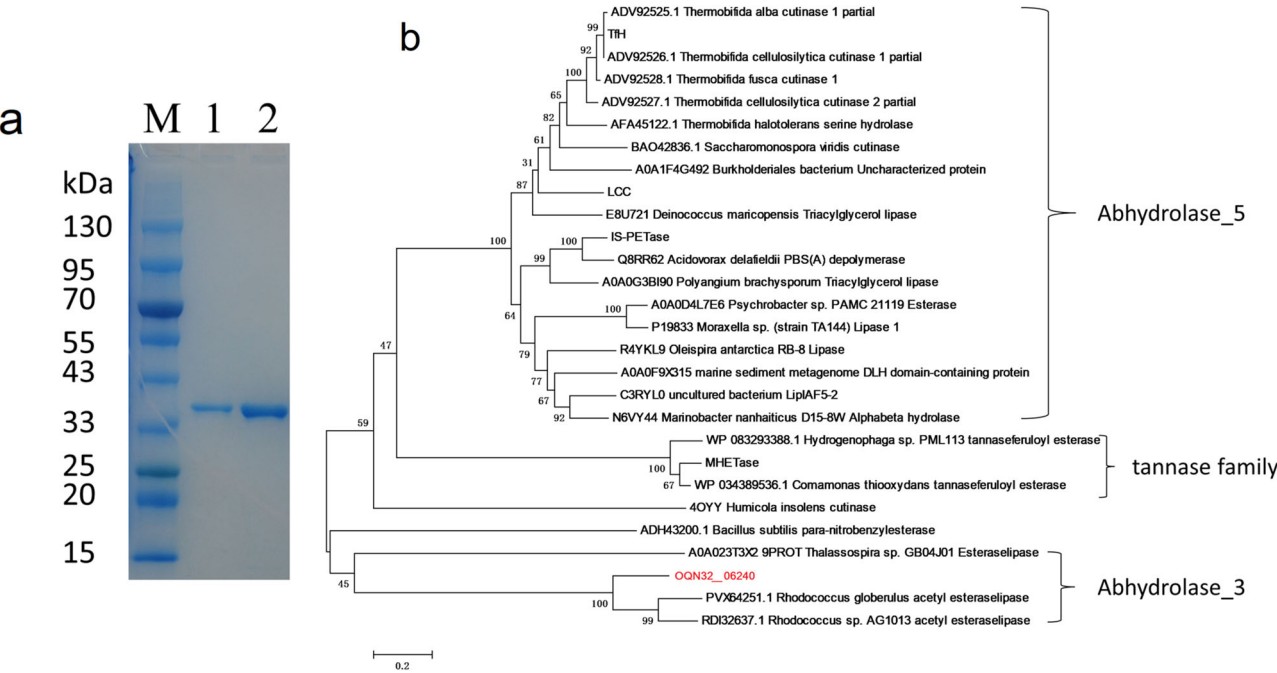

**Fig. 3 PET esterase (OQN32_06240) of *R. pyridinivorans* P23. a** Purification of the hexa-histidine-tagged PET esterase (OQN32_06240), 34 kDa.
**b** Phylogenetic tree of PET esterase (OQN32_06240) with known PET hydrolytic enzymes and MHET hydrolytic enzymes. The GenBank or Protein Data Bank accession numbers with the organism source of proteins are shown on the leaves. Bootstrap values are shown at the branch points. Scale bar, 0.2 amino acid substitutions per single site.

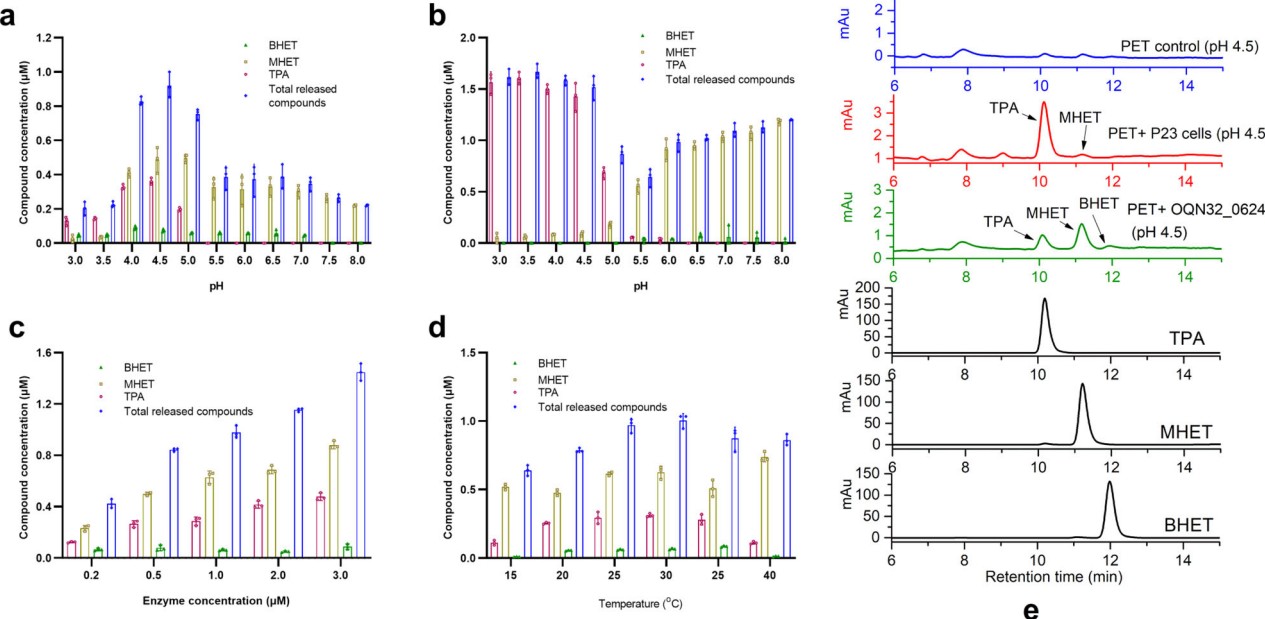

**Fig. 4 PET esterase (OQN32_06240) of *R. pyridinivorans* P23 towards PET. a**, **b** Enzymatic degradation of PET by 1.0 μM purified PET esterase (OQN32_06240) (**a**) or cells of *R. pyridinivorans* P23 (OD$_{600}$ = 1.0) (**b**) in pH 3.0–8.0 Na$_2$HPO$_4$-NaH$_2$PO$_4$ buffer. $n$ = 3 biologically independent experiments were performed. **c**, **d** Effect of enzyme concentration (**c**) and temperature (**d**) on enzymatic PET film hydrolysis in pH 4.5 Na$_2$HPO$_4$-NaH$_2$PO$_4$ buffer. $n$ = 3 biologically independent experiments were performed. **e** High-performance liquid chromatography spectra of the products released from the PET film with *R. pyridinivorans* P23 cells and purified PET esterase (OQN32_06240). The PET film in pH 4.5 buffer without cells or purified enzyme was set as blank control. TPA, MHET and BHET standards were also set as references. Purple circle for TPA, the yellow square for MHET, green triangle for BHET and blue diamond for total released compounds. Error bars, standard deviation (s.d.).

than 1.5 µM compounds (TPA and MHET) released, of which more than 94.0% was TPA (Fig. 4b, e). In pH 3.0–5.0 enzymatic degradation solution, the released compounds were found to be mainly TPA, while more than 87.2% of MHET was detected in pH 5.5–8.0 enzymatic degradation solutions (Fig. 4b). At pH 4.5 and 30 °C, with the increase of purified enzyme concentration from 0.2 to 3.0 µM, the totally released compounds also increased from 0.42 to 1.41 µM (Fig. 4c). The purified PET esterase (OQN32_06240) exhibited relatively high enzymatic activity at 25–30 °C and best at 30 °C (Fig. 4d). It could be speculated from these results that PET esterase (OQN32_06240) first degraded PET into mainly MHET with a small amount of BHET, then hydrolyzed MHET into TPA in acidic conditions (pH 3.0–5.0).

**Characterization of PET esterase (OQN32_06240) towards BHET.** To verify our speculation, PET esterase (OQN32_06240) from heterologous expression in *E. coli* and whole-cell biocatalyst anchoring on the cell surface were both used to catalyze BHET. We tested the BHET degradation activity of purified PET esterase (OQN32_06240) from *E. coli* with 1.0 mM BHET as substrate in the pH 3.0–8.0 buffers at 30 °C. Results indicated that it had relatively high enzymatic activity at pH 3.5–7.5 and best activity at pH 7.0, with 40.7% BHET transformed to MHET in 48 h (Fig. 5a, g). When cells of *R. pyridinivorans* P23 were used as whole-cell biocatalyst ($OD_{600}$ of 1.0) towards 1.0 mM BHET at 30 °C in pH 3.0–8.0 buffers, 27.2–56.4% BHET were transformed into MHET in 30 min (Fig. 5b, g). At this time, as BHET was the main component in the reaction solution, PET esterase was occupied by BHET, and MHET could hardly be catalyzed into TPA. As the reaction time extended to 16 h, almost all BHET were transformed into MHET and then TPA. TPA in the enzymatic supernatant accounts for 5.25–49.05% in pH 3.0–6.0 buffers, with the highest proportion of 49.05% at pH 4.5 (Fig. 5c, g). In contrast, only 0.3–1.4% TPA could be detected in pH 6.5–8.0 buffers (Fig. 5c). The $K_{cat}/K_m$ value of purified PET esterase (OQN32_06240) was determined to be $1.63 \, M^{-1} \, S^{-1}$ towards BHET at pH 4.5, 30 °C (Supplementary Fig. 10a).

**PET esterase (OQN32_06240) degrades MHET to TPA in the acidic environment.** We also determined the enzymatic activity of PET esterase (OQN32_06240) towards MHET. When 1.0 mM MHET was served as the substrate of 5.0 µM PET esterase (OQN32_06240) at 30 °C, results indicated that it had relatively high enzymatic activity in pH 3.5–5.0 with 2.2–3.6% of MHET degraded and highest enzymatic activity at pH 4.5 with 3.6% MHET transformed to TPA in 48 h (Fig. 5d, h). Besides, less than 1.0% of MHET was degraded into TPA in the pH 3.0 and 5.5–8.0 (Fig. 5d). The same profile was observed when cells of *R. pyridinivorans* P23 were used as whole-cell biocatalyst ($OD_{600}$ 1.0) and 1.0 mM MHET was served as the substrate at 30 °C. When the reactions were performed for 5 h, 20.0–23.7% of MHET was transformed into TPA in pH 3.5–5.0, in which conditions were suitable for PET esterase (OQN32_06240) to degrade MHET (Fig. 5e, h). *R. pyridinivorans* P23 cells exhibited relatively high enzymatic activity at pH 3.0–5.5 with 16.9–49.3% of MHET degraded and highest enzymatic activity at pH 4.5 with 49.3% MHET transformed to TPA in 16 h (Fig. 5f, h). Predictably, less than 5.0% of MHET was degraded into TPA in pH 6.0–8.0 (Fig. 5f). The difference between the enzymes anchoring on the cell membrane and the free-moving enzymes is that the anchoring one can withstand a more acidic environment, and work in lower pH conditions. The $K_{cat}/K_m$ value of purified PET esterase (OQN32_06240) was determined to be $0.102 \, M^{-1} \, S^{-1}$ towards MHET at pH 4.5, 30 °C (Supplementary Fig. 10b).

**PET degradation pathway in *R. pyridinivorans* P23.** The processes of PET degradation, TPA transportation, and degradation in *R. pyridinivorans* P23 were illustrated in Fig. 6a. In *R. pyridinivorans* P23, cell membrane anchoring PET esterase encoded by a chromosome gene with the locus of OQN32_06240 first hydrolyzes PET to produce MHET (the primary product) and tiny BHET. Then, PET esterase (OQN32_06240) hydrolyzes MHET to TPA and EG further, which is more effective under acidic conditions. In *R. pyridinivorans* P23, TPA is incorporated through the TPA transporter (PcaK, OQN32_25295) and catabolized by TPA 1,2-dioxygenase (TPADO, OQN32_25275, OQN32_25280, OQN32_25290), followed by 1,2-dihydroxy-3,5-cyclohexadiene-1,4-dicarboxylate dehydrogenase (DCDDH, OQN32_25285) to produce protocatechuic acid (PCA) (Supplementary Table 2, Supplementary Fig. 11). These five genes and gene OQN32_25270 encoding a regulatory protein PcaR are clustered on the pA plasmid. The resultant PCA is ring-cleaved by PCA 3,4-dioxygenase (Pca34, OQN32_09320, OQN32_09325) and further catabolized by enzymes arranged in another operon (OQN32_22690-OQN32_22715) located on the chromosome (Supplementary Table 2, Supplementary Fig. 11).

**Differential RNA-seq analysis of TPA/PET degradation pathway enzymes of cells derived from different culture conditions.** The gene transcription level of PET esterase (OQN32_06240) was similar under TPA and 2216 marine broth culture conditions but lower in PET culture condition as the cell growth was relatively slow under this condition (Supplementary Table 3). These results indicated that PET esterase (OQN32_06240) was a constructive expression enzyme and not inducible by PET. The mRNA abundance of the genes involved in the TPA degradation pathway (OQN32_25270-OQN32_25295) increased in both TPA and PET cultured conditions, especially when 5.0 mM of TPA was served as the substrate, the genes' transcription level increased more significantly (Supplementary Table 3). For instance, the transcriptional level of *pcaK* encoding an MFS transporter, which was proposed to serve as a TPA transporter in *R. pyridinivorans* P23, was up-regulated from RPKM value of 3.94 in 2216 marine broth culture condition to RPKM values of 15.53 and 16,043.44 in PET and TPA culture conditions, respectively (Supplementary Table 3). This indicated that TPA was released from PET into the culture supernatant to induce high transcription level of TPA degradation pathway-related genes when *R. pyridinivorans* P23 was cultured with PET. As we mentioned in Fig. 2b, in the PET film cultivation, a total of 4.03 mg weight loss of PET indicated only 0.14 mM TPA and EG each were released gradually into the culture medium in the culture period as long as five weeks. Besides, the TPA transportation and consumption was accompanying its production, which resulted in a low concentration of TPA in the culture medium during the long culture process. As a result, the increased fold change of the transcription level of the genes involved in the TPA degradation pathway was relatively low in the PET cultured condition, compared with the TPA culture condition. The transcription level of genes involved in the PCA degradation pathway was also up-regulated under PET and TPA culture conditions (Supplementary Table 3).

**The biodiversity of bacteria in the environment present the same PET degradation profile with *R. pyridinivorans* P23.** Bacteria with the coexistence of PET esterase and the large subunit of the oxygenase component of TPA 1,2-dioxygenase (TPADO) homologs might possess the same PET degradation profile with *R. pyridinivorans* P23. In the Pfam search pattern, a total of 7558 genome hits were found with both PET esterase (PF07859) and the large subunit of the oxygenase component of

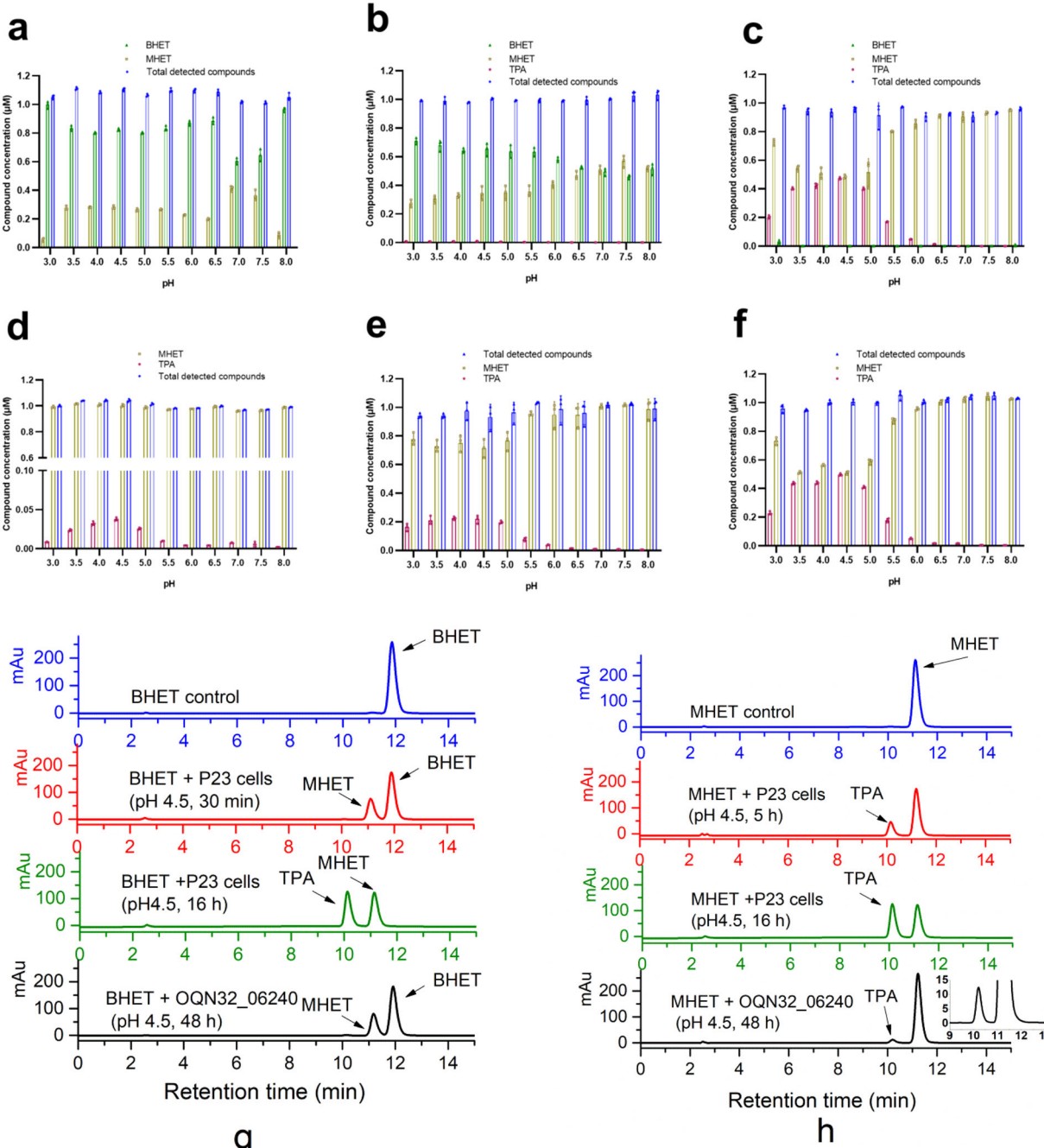

**Fig. 5 PET esterase (OQN32_06240) of *R. pyridinivorans* P23 towards BHET and MHET. a–c** Enzymatic degradation of 1.0 mM BHET in pH 3.0–8.0 $Na_2HPO_4$-$NaH_2PO_4$ buffer by 1.0 μM purified PET esterase (OQN32_06240) for 48 h (**a**), or *R. pyridinivorans* P23 cells (OD$_{600}$ = 1.0) for 30 min (**b**) and 16 h (**c**), respectively. *n* = 3 biologically independent experiments were performed. **d–f** Enzymatic degradation of 1.0 mM MHET in pH 3.0–8.0 $Na_2HPO_4$-$NaH_2PO_4$ buffer by 5.0 μM purified PET esterase (OQN32_06240) for 48 h (**d**), or *R. pyridinivorans* P23 cells (OD$_{600}$ = 1.0) for 5 h (**e**) and 16 h (**f**) respectively. *n* = 3 biologically independent experiments were performed. **g** HPLC spectra of BHET degradation by purified PET esterase (OQN32_06240) for 48 h or by *R. pyridinivorans* P23 cells in pH 4.5 for 30 min and 16 h respectively. **h** HPLC spectra of MHET degradation by purified PET esterase (OQN32_06240) for 48 h or by *R. pyridinivorans* P23 cells in pH 4.5 for 5 h and 16 h, respectively. Purple circle for TPA, yellow square for MHET, green triangle for BHET and blue diamond for total detected compounds. Error bars, standard deviation (s.d.).

TPADO (PF00848) homologs (Supplementary Fig. 12). In the phylum level, 4,900 genomes belong to *Proteobacteria*, accounting for the largest proportion of 64.83%, and 1717 genomes (22.72%) belong to *Actinobacteriota* to which *R. pyridinivorans* P23 belongs (Supplementary Fig. 12). In addition to *Proteobacteria* and *Actinobacteriota*, *Bacteroidota* (208, 2.75%), *Firmicutes* (155, 2.05%), *Acidobacteriota* (105, 1.39%), *Planctomycetota* (83, 1.10%) are the other four phyla with proportion of genome hits more than 1.0%

(Supplementary Fig. 12). In the KEGG search pattern, a total of 895 to 52 genome hits were found with 30% to 100% identities with *Proteobacteria* and *Actinobacteriota* as the main compositions (Fig. 7a). Detailed analysis of 321 genome hits (60% identity) in the genus level was performed. In the genus level, 51 hit genomes belong to *Burkholderia*, accounting for the largest proportion of 15.89%, and seven hit genomes (2.18%) belong to *Rhodococcus*, to which *R. pyridinivorans* P23 belongs (Fig. 7b).

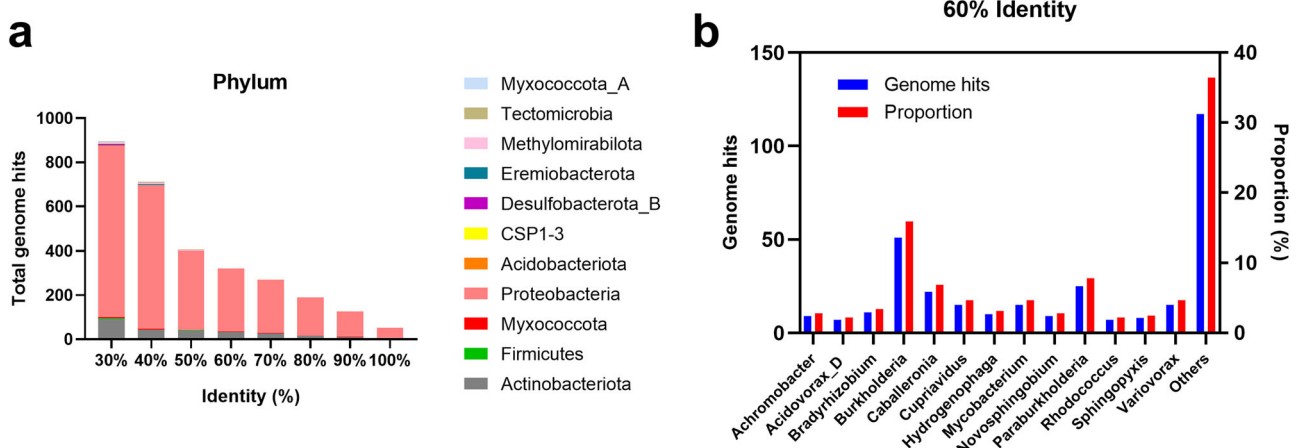

**Fig. 6 Detailed description of PET degradation processes by *R. pyridinivorans* P23. a** Proposed PET degradation and assimilation pathway in *R. pyridinivorans* P23. *R. pyridinivorans* P23 cell surface displayed PET esterase (OQN32_06240) first degrades PET into mainly MHET with a small amount of BHET, and then hydrolyzes MHET into TPA eventually for transportation and assimilation. **b** The proposed PET biodegradation model with biofilm formation in the marine environments. Step 1, adhesion to the surface of PET; step 2, cell surface displayed PET esterase (OQN32_06240) hydrolyzes PET into mainly MHET, which makes the microenvironment acidic; step 3, MHET hydrolase activity of PET esterase (OQN32_06240) activated in acidic environment and degradation of MHET to produce TPA; step 4, TPA transporter induction, transport and degradation of TPA and propagation of bacteria.

**Fig. 7 Biodiversity of PET degrading bacteria in the environment having the same PET degradation pathway with *R. pyridinivorans* P23 using KEGG pattern. a** Genome hits (30–100% identity) with the coexistence of PET esterase (K01066) and large subunit of the oxygenase component of TPADO (K16319) in the phylum level showing *Proteobacteria* and *Actinobacteriota* as the main compositions. **b** Bacteria composition in genus level with 60% identity of genome hits showing *Burkholderia* accounting for the largest proportion 15.89%.

The other hit genomes belong to *Paraburkholderia* (25, 7.79%), *Caballeronia* (22, 6.85%), *Mycobacterium* (15, 4.67%), *Cupriavidus* (15, 4.67%), *Variovorax* (15, 4.67%), *Bradyrhizobium* (11, 3.43%), *Hydrogenophaga* (10, 3.12%), *Novosphingobium* (9, 2.80%), *Achromobacter* (9, 2.80%), *Sphingopyxis* (8, 2.49%), *Acidovorax_D* (7, 2.18%) and others (117, 36.45%) (Fig. 7b). These results indicated that environmental bacteria possessing PET esterase couple with TPA degradation pathway are frequently found in environmental microorganisms.

## Discussion

Based on the results, we proposed a model for PET biodegradation by *R. pyridinivorans* P23 in the marine environment, in which *R. pyridinivorans* P23 attached to PET surface forming biofilm (Fig. 6b). As the PET esterase (OQN32_06240) is anchoring on the cell surface of *R. pyridinivorans* P23, free-living marine *R. pyridinivorans* P23 cells have to attach to the PET surface for the biodegradation of PET (Fig. 6b, step 1). After the attachment of *R. pyridinivorans* P23 to PET, the constitutive expression of PET esterase (OQN32_06240) hydrolyzed PET, producing mainly MHET, which would be retained in the biofilm on the PET surface (Fig. 6b, step 2). With the accumulation of MHET, the microenvironment between the biofilm and PET surface became acidic due to the carboxyl group of MHET. The MHET hydrolysis activity of PET esterase (OQN32_06240) was then activated, which contributed to the hydrolysis of MHET and the accumulation of TPA, causing the microenvironment between biofilm and PET surface to become more acidic (Fig. 6b, step 3). Then, the TPA transporter would be induced to express due to the accumulation of TPA, which would be transported into cells for complete oxidation to provide energy and carbon source for biofilm propagation (Fig. 6b step 4). In addition to the TPA transporter, other genes involved in the TPA degradation pathway would be induced in biofilm, according to the preliminary RNA-seq results of flask cultivation of *R. pyridinivorans* P23 with PET and TPA (Supplementary Table 3), and contribute to the utilization of TPA. In this model, the characteristics of PET esterase (OQN32_06240) including cell surface anchoring, constitutive expression, and PET, BHET and MHET (in acid environment) hydrolysis activities were all confirmed in this study. The retention of MHET and TPA by biofilms results in creating an acidic microenvironment was speculated based on the MHET hydrolysis activity of PET esterase (OQN32_06240) in acid environment. As MHET was produced from the PET surface under the coverage of biofilm, steep gradients of MHET concentration and pH were produced in the biofilm, which caused the heterogeneous physiological activity of the biofilm. The cells close to the plastic surface in the biofilm are easier to obtain TPA for proliferation, while the cells far away from the plastic surface gradually become extinct due to less acquiring of TPA. This can ensure the cells in the biofilm closer to the plastic have the vitality to provide PET degrading enzymes to decompose PET for the biofilm propagation on PET. In this model, biofilm plays two crucial roles in the biodegradation of PET. Biofilm will retain the released products, prevent them from losing in the ocean, and ensure the carbon source supply of microorganisms. Besides, the biofilm will separate the acid microenvironment on the plastic surface under its coverage from the alkaline seawater, making sure the activation of MHET hydrolase activity of PET esterase (OQN32_06240). In the flask cultivation of *R. pyridinivorans* P23 with PET, the biodegraded products would be released into the culture supernatant for acquiring by cells. However, in marine conditions, the biofilm formation on plastic is a crucial and smart strategy to retain the released products.

Biofilm is one of the Earth's most widely distributed and successful modes of life[42]. The bacterial colonization and biofilm formation on different plastic surfaces, including low-density polyethylene (LDPE), PS, and PET, has been widely reported[43,44]. As plastics and biofilms have a complicated relationship, the precise mechanism of biofilm to degrade plastic for its proliferation is still unclear until now. In marine environments, PET not only provides a surface for attachment, colonization, and biofilm formation of marine bacteria, but also a carbon source for PET-degrading marine microorganisms. This study clarified the formation and survival mechanism of *R. pyridinivorans* P23 biofilm on PET surface. In marine environments, the community composition of biofilm on plastic is certainly much more complicated, and *R. pyridinivorans* P23 definitely plays a crucial role in the biofilm community.

PET esterase (OQN32_06240) hydrolyzes PET to produce MHET (the primary product) and tiny BHET, which is similar to that of PETase from *Ideonella sakaiensis* 201-F6 (*Is*-PETase)[18]. Compared with the typical *Is*-PETase, PET esterase of *R. pyridinivorans* P23 has several distinct characteristics. First, in *R. pyridinivorans* P23, PET esterase is a cell membrane anchoring protein displaying on the cell surface, while the *Is*-PETase is a secretory one released into the environment[18]. The difference between the existence patterns of these two PET hydrolase is presumed to be closely related to the isolation habitats of the two strains. *Ideonella sakaiensis* 201-F6 was isolated from PET bottle recycling site[18], while *R. pyridinivorans* P23 was from a marine sediment sample. The marine environment requires that the PET-degrading enzyme of marine microorganisms cannot be secreted; otherwise, the enzyme will be lost after being secreted and cannot be utilized to degrade PET for the survival of themselves. Secondly, *Is*-PETase is inducible, while PET esterase of *R. pyridinivorans* P23 is in a low and constitutive expression mode, which could be attributed to the spatial limitations of the cell membrane on which it anchors. Thirdly, although most PET hydrolases were reported to have MHET hydrolase activity, none was activated under an acid environment like PET esterase of *R. pyridinivorans* P23[18]. The MHET hydrolase activity of PET esterase from *R. pyridinivorans* P23 is generally inactive in seawater (about pH 8). Thus, the biofilm formation on the PET surface by *R. pyridinivorans* P23 would make the microenvironment between the surfaces of PET and biofilm relatively cut off from alkaline seawater. With the gradual release of MHET from PET into the microenvironment under the catalysis of PET esterase from *R. pyridinivorans* P23, the microenvironment would turn acid gradually, being suitable for MHET hydrolysis by PET esterase.

PET esterase (OQN32_06240) presents in the phylogenetic tree on a branch quite far away from known PET hydrolytic enzymes including *Is*-PETase (Abhydrolase_5) (Fig. 3b). In this study, PET esterase (OQN32_06240) was identified based on PET degrading strain isolation followed by active PET hydrolase tracking and identification, which was more likely to discover enzymes with distinct sequences. Thus, PET esterase (OQN32_06240) would provide an enzyme sequence for modification to improve the enzyme activity and properties in the future. In this study, PET esterase (OQN32_06240) with 0.92 µM products released from PET showed significantly lower enzymatic activity than *Is*-PETase with 0.3 mM products released[18]. The enzymatic activity of PET esterase (OQN32_06240) towards PET was also lower than Ple628 with 52.9 µM MHET released[45], and lower than PE-H from a marine *Pseudomonas* strain with 4.2 mg/L (20 µM) MHET released[46]. *E. coli* BL21(DE3) may not be a suitable chassis for the production of Gram-positive bacterium-originated PET esterase (OQN32_06240) as the relatively low enzyme activity obtained. However, cell anchoring PET esterase (OQN32_06240) of *R. pyridinivorans* P23 could be easily prepared from the collection of cells after fermentation and used as a cell biocatalyst to degrade PET, rendering it a promising way of PET degradation.

In this study, 4.28% PET weight loss was observed in 5 weeks cultivation of *R. pyridinivorans* P23 which was comparable with 1.3–1.8% of PET weight loss after 30 days incubation with four marine strains *Marinobacter sediminum* BC31_3_A1, *Marinobacter gudaonensis* BC06_2_A6, *Thalassospira xiamenensis* BC02_2_A1 and *Nocardioides marinus* BC14_2_R3[21]. Meanwhile, PET weight loss by *R. pyridinivorans* P23 (4.28%) was significantly lower than that by *I. sakaiensis* 201-F6, with 60 mg of PET film almost completely degraded in 6 weeks of cultivation[18]. Although the efficiency of *R. pyridinivorans* P23 in degrading PET currently does not reach the level of industrial application, as it was discovered in the marine environment, which is the gathering place of discarded plastic waste in the environment, this bacterium and its similar PET degradation and assimilation functional microbial groups will play an essential role in the natural bioremediation of marine plastic pollution.

Although a few PET-degrading strains have already been reported, the biodiversity of bacteria with PET degradation and assimilation capacity in the environment, especially the ocean, still needs to be discovered. In this study, *Proteobacteria* and *Actinobacteriota* were the main phyla with the presence of PET esterase and the large subunit of the oxygenase component of TPADO. Evidence has been provided in another investigation that the PET hydrolases originate mainly from phyla *Actinobacteria*, *Proteobacteria*, and *Bacteroidetes*[32]. The restriction of PET hydrolases and PET assimilation capability to a few bacterial phyla indicate that this metabolic capability has evolved for a short time since the production and release of PET into the environment and thus limited to a very few phylogenetic groups. It is speculated that the groups of *Proteobacteria* and *Actinobacteriota* with TPA degradation capability are more likely to survive with PET as the carbon and energy source as it could acquire PET hydrolase activity from the evolution of its native esterase.

## Materials and methods

**Screening and identification of PET degrading strain**. The Bacterial strain used in this study was isolated from a marine sediment sample of the Pacific Ocean. Approximately 1.0 g of the sediment sample was inoculated into 150 ml mineral salt medium with 1.0 g of poly(ethylene terephthalate) (PET) powder (crushed from granular PET, 429252, Sigma-Aldrich, USA) as the sole carbon and energy source and cultivated at 30 °C, rotation speed 200 rpm for one month for the enrichment of PET degrading bacteria. The mineral salt medium contains (per liter of water): 1.0 g $KH_2PO_4$, 10.0 g $(NH_4)_2SO_4$, 0.4 g $MgSO_4 \cdot 7H_2O$, 0.5 g NaCl, 2.5 g carbamide, 0.0001 g $FeSO_4 \cdot 7H_2O$, 0.0001 g $MnSO_4 \cdot 4H_2O$, and 50 μg Biotin, pH 7.0. After enrichment, 100 μl of the culture was spread on 2216 marine agar (Haibo Corporation, Qingdao, China) plate. The 2216 marine agar medium contains (per liter of water): 5.0 g peptone, 1.0 g yeast extract, 0.10 g ferric citrate, 19.45 g NaCl, 5.90 g $MgCl_2$, 3.24 g $Na_2SO_4$, 1.80 g $CaCl_2$, 0.55 g KCl, 0.16 g $Na_2CO_3$, 0.08 g KBr, 34.00 mg $SrCl_2$, 22.00 mg $H_3BO_3$, 4.00 mg $Na2SiO3 \cdot 9H_2O$, 2.40 mg NaF, 1.60 mg $Na_2NO_3$, 8.00 mg $Na_2HPO_4$, and 15.0 g agar, pH 7.6. Colonies were inoculated into a test tube containing 5 ml of mineral salt medium with a piece of PET film (Good Fellow GF89357619) as the sole carbon and energy source and cultivated at 30 °C, 200 rpm for three days. PET degradation activity was estimated visually based on the change of transparency of the culture medium due to the growth of cells. Strain P23 was observed to grow in mineral salt medium with PET film, selected and identified using 16S rDNA sequence, obtained from whole genome sequencing. A phylogenetic tree was constructed using the Mega 6.0 program using the Neighbor-joining method based on the 16S rRNA gene.

**Degradation of PET film by *R. pyridinivorans* P23**. A single colony of *R. pyridinivorans* P23 from the 2216 marine agar plate was inoculated into a test tube (∅18× 180 mm) containing 5 ml 2216 marine broth medium (pH 7.0) and shaken at 200 rpm, 30 °C overnight. Then, 5 ml of bacterial culture was centrifuged to remove the remaining medium. The precipitate was washed three times with 1.0 ml 20 mM phosphate buffer and inoculated into a 500 ml flask containing 150 ml of mineral salt medium with a piece of 8.0 cm × 8.0 cm PET film (Good Fellow GF89357619, thickness 0.013 mm) weighing 92.9, 93.7, and 96.2 mg respectively for three replicates as sole carbon and energy source. Cultivations of PET film in mineral salt medium without inoculation were set as negative controls. After cultivation under 200 rpm at 30 °C for seven days, the remaining PET film was removed, washed, dried, weighed, and served for the next round of seven days of cultivation for a total of five weeks. In order to observe the growth of *R. pyridinivorans* P23 on PET film, 20 μl of the prepared inoculum was spread on a mineral salt medium agar plate (pH 7.0, 1.5% agar) and covered with a piece of sterile 1.0 cm× 1.5 cm PET film (Good Fellow GF89357619, thickness 0.013 mm) at 30 °C for seven days. After the formation of biofilm on it, the PET film was removed from the agar plate with tweezers and directly subjected to environmental scanning electron microscope (SEM, FEI Quanta450, USA) observation of *R. pyridinivorans* P23 cells grown on it.

**Determination of substance released by PET degradation**. Culture fluid (150 ml) of *R. pyridinivorans* P23 grown on PET film and control cultures without inoculation were collected at 8 h, 30 h, and 7 d, respectively, and centrifuged (6,000 × g, 10 min, 4 °C) to remove the cells. The supernatant was adjusted to pH 2.0 using 2.0 M HCl and extracted with 150 mL of ethyl acetate twice. The ethyl acetate layer was recovered and evaporated on a rotary evaporator (Buchi, Switzerland). The residue was dissolved in 1.0 ml methanol, filtered through a 0.22 μm pore size filter (Shimadzu, Japan), and subjected to reverse-phase high-performance liquid chromatography (HPLC) analysis. HPLC was performed on a 1260 infinity system (Agilent Technologies, USA) equipped with a Cosmosil $5C_{18}$-AR-II analytical column (4.6 × 250 mm) from Nacalai Tesque (Kyoto, Japan). The mobile phase was 35% methanol/ 65% 20 mM phosphate buffer (pH 1.5) at a flow rate of 1.0 ml/min, and the effluent was monitored at a wavelength of 240 nm. Bis(2-hydroxyethyl) terephthalic acid (BHET, Sigma-Aldrich, USA), mono(2-hydroxyethyl) terephthalic acid (MHET, Amatek Scientific, China), and disodium terephthalate (TPA-$Na_2$, Aladdin, China) were dissolved in methanol and used as standards.

**Complete genome sequence analysis of *R. pyridinivorans* P23**. *R. pyridinivorans* P23 was grown in 2216 marine broth medium at 30 °C, and its genomic DNA was extracted and purified using a bacteria genomic DNA kit (Tiangen, Beijing, China) according to the manufacturer's instructions. The complete genome of *R. pyridinivorans* P23 was constructed based on the co-assembly of short reads (150 bp × 2) and long reads (average 4,029.55 bp) derived from the Illumina HiSeq sequencing platform and third-generation single-molecule sequencing technology with Pac-Bio RSII sequencer conducted at MajorBio Technologies (Shanghai, China)[47]. A total of 1,014,860,906 bp and 2,263,058,586 bp high-quality data derived from the Illumina platform and third-generation platform, respectively, were obtained and assembled into a circular chromosome and four plasmids using the hierarchical genome assembly process (HGAP) algorithm[48,49]. The open reading frames were predicted by Prodigal[50], and the functions were annotated using BLAST searches of non-

redundant protein sequences from the COG[51], GO[52], KEGG[53], Swiss-Prot[54], and NCBI-nr databases[55].

**RNA purification and RNA-Seq.** *R. pyridinivorans* P23 was cultured in mineral salt medium with a piece of 8.0 cm× 8.0 cm PET film or 1 g/l TPA as the sole carbon source respectively, or 2216 marine broth medium at 30 °C (Fig. 2a, Supplementary Fig. 3c, d). Cells of 100 ml culture in the late exponential phase of growth (6 d for PET, 16 h for 2216 marine broth medium, and 12 h for TPA) were harvested by centrifugation at 6000 × g, and 4 °C and then the centrifugation precipitates of the cultures were washed three times with ice-cooled $H_2O$. Total RNA was isolated from cells using the Trizol Reagent (Invitrogen, Carlsbad, CA, USA) according to the manufacturer's instructions. Two biological replicates each were used in the preparation of RNA samples from the three different substrate culture conditions. RNA sample preparation, RNA concentration, and purity determination were carried out according to routine methods[56]. RNA-Seq performed on Illumina HiSeq™ 2000 platform and subsequent bioinformatics analysis was carried out by MajorBio Technologies (Shanghai, China) (Supplementary Table 4). Differentially transcribed genes were identified using the MajorBio platform (https://cloud.majorbio.com/) (Supplementary Data 1). The RPKM (Reads Per Kilobase per Million mapped reads) method was used to calculate unigene transcription and directly compare gene transcription levels between samples with different cultivation conditions[57].

**Heterologous expression of protein PET esterase (OQN32_06240).** The gene for PET esterase (OQN32_06240) was commercially synthesized with codon optimization for expression in *Escherichia coli* BL21 (DE3) (Sangon, Shanghai, China, Supplementary materials). The amino acid sequence corresponding to the transmembrane domain was predicted using the TMHMM server V2.0 (http://www.cbs.dtu.dk/services/TMHMM/) and removed from the synthetic DNA. The synthetic gene for PET esterase (OQN32_06240) was cloned into the *Nde*I (5′ end) and *Xho*I (3′ end) sites of pET-30a vector (Sangon, Shanghai, China) through fusion to its C-terminal hexa-histidine tag-coding sequence. PET esterase (OQN32_06240) gene was expressed in *E. coli* BL21 (DE3) by 0.5 mM IPTG induction at 20 °C for 16 h. The *E. coli* cells were harvested by centrifugation (4000 × g, 10 min, 4 °C), resuspended in lysis buffer (50 mM Tris-HCl, 300 mM NaCl, 0.1% Triton X-100, pH 8.0) and disrupted by sonication on ice. After centrifugation (12,000 × g, 15 min, 4 °C), insoluble debris was removed, and the supernatant containing the soluble protein was transferred to a Ni-affinity chromatography column (GE Healthcare) pre-equilibrated with binding buffer (50 mM Tris-HCl, 300 mM NaCl, pH 8.0). After washing unbound proteins with the washing buffer (50 mM Tris-HCl, 300 mM NaCl, 20/50 mM imidazole, pH 8.0), the bound proteins were eluted with elution buffer (50 mM Tris-HCl, 300 mM NaCl, 500 mM imidazole, pH 8.0). Fractions containing the recombinant protein were collected and dialyzed at 4℃ against binding buffer (50 mM Tris-HCl, 300 mM NaCl, pH 8.0) to remove the imidazole. The recombinant protein was then concentrated by ultrafiltration with centrifugal filters (Ultracel-3 k, Merck Millipore, IRL) and subjected to sodium dodecyl sulfate-polyacrylamide gel electrophoresis (SDS-PAGE, 12% polyacrylamide) for visual estimation of molecular mass and purity. The purified protein concentration was determined with a Non-Interference Protein Assay Kit (Order NO. C503071, Sangon, Shanghai, China).

**Enzyme assays for PET film.** Purified protein (1.0 µM) of PET esterase (OQN32_06240) was incubated with 4 pieces of PET film (1.0 cm × 1.0 cm × 0.013 mm, Good Fellow GF89357619) in 300 µl buffer containing 300 mM $Na_2HPO_4$-$NaH_2PO_4$ (pH 3.0–8.0) at 30 °C for 60 h. The reaction was terminated by heat treatment (80 °C, 10 min). After the remaining PET film was removed, the supernatant obtained was filtered through a 0.22 µm filter and analyzed by HPLC. Reactions without the addition of purified protein were used as negative controls. To investigate the impact of temperature on the biodegradation of PET film, reactions were carried out in 300 mM $Na_2HPO_4$-$NaH_2PO_4$, pH 4.5 with 1.0 µM purified protein at 15–35 °C, respectively, for 60 h. In the investigation of enzyme concentration on the biodegradation of PET film, 0.2–3.0 µM of purified protein was used in reactions which were carried out in 300 mM $Na_2HPO_4$-$NaH_2PO_4$, pH 4.5 at 30 °C for 60 h.

**Enzyme assays for BHET and MHET.** The purified protein (1.0 µM) was incubated with 1.0 mM BHET (Sigma-Aldrich, USA) in 300 µl buffer containing 300 mM $Na_2HPO_4$-$NaH_2PO_4$, pH 3.0–8.0 at 30 °C for 48 h. Besides, the purified protein (5.0 µM) was incubated with 1.0 mM MHET (Amatek Scientific, China) in 300 µl buffer containing 300 mM $Na_2HPO_4$-$NaH_2PO_4$, pH 3.0–8.0 at 30 °C for 48 h. After reactions, the supernatant was immediately filtered through a 0.22 µm filter and analyzed by HPLC.

**Kinetic analysis of PET esterase (OQN32_06240) towards BHET and MHET.** One µM purified protein was incubated with 0.1 mM to 5.0 mM BHET in 300 mM $Na_2HPO_4$-$NaH_2PO_4$, pH 4.5 at 30 °C. Besides, 1.0 µM purified protein was incubated with 0.2 mM to 3.0 mM MHET in 300 mM $Na_2HPO_4$-$NaH_2PO_4$, pH 4.5 at 30 °C. After the reaction, the supernatant was immediately filtered through a 0.22 µm filter and analyzed by HPLC for MHET and TPA produced. Initial degradation rates of PET esterase (OQN32_06240) against BHET and MHET were plotted against BHET and MHET concentrations, respectively, and the kinetic parameters were determined by using the Michaelis-Menten equation utilizing Graph Pad Prism version 6.01 (GraphPad Software, San Diego, CA).

**PET, BHET, and MHET degradation by cells of *R. pyridinivorans* P23.** Cells of *R. pyridinivorans* P23 was used as whole-cell biocatalyst towards PET film and incubated with four pieces of PET film (1.0 cm × 1.0 cm × 0.013 mm, Good Fellow GF89357619) to final concentration of $OD_{600}$ 1.0 in a 1.5 ml centrifuge tube containing 300 µl 300 mM $Na_2HPO_4$-$NaH_2PO_4$, pH 3.0–8.0 at 30 °C for 60 h. The tube was shaken at 100 rpm to prevent cell precipitation. When towards BHET and MHET degradation, 1.0 mM BHET or 1.0 mM MHET was used as substrate under the same reaction conditions with PET film. The reaction time was 30 min and 16 h for BHET, and 5 h and 16 h for MHET, respectively, to make sure the dynamic transformation of BHET to MHET and then TPA could be observed. The supernatant obtained was filtered through a 0.22 µm filter analyzed by HPLC.

**Protein homolog search on the fully sequenced genomes.** The Annotree server (http://annotree.uwaterloo.ca/app/) was utilized to search the coexistence of PET esterase (PF07859, K01066) and large subunit of the oxygenase component of TPADO (PF00848, K16319) homologs in bacteria[58]. In the Pfam search pattern, homologs of PF07859 and PF00848 with ≤1e$^{-5}$ *E*-value were searched against the genome database implemented in this server and displayed at the phylum level. In the KEGG search pattern, homologs of K01066 and K16319 with 30–100% identity, ≤1e$^{-5}$ *E*-value were searched against the genome database and displayed

at the phylum level. Detailed analysis of genome hits of 60% identity in genus level was performed to further reveal bacteria involved in PET degradation like *R. pyridinivorans* P23.

**Statistical analysis**. Error bars indicate the standard error of the mean of the three independent experiments.

**Reporting summary**. Further information on research design is available in the Nature Portfolio Reporting Summary linked to this article.

## Data availability

The complete genome sequence of *R. pyridinivorans* P23 was deposited in the GenBank database under the accession number CP113798-CP113802. The RNA-Seq reads have been deposited in GenBank with accession numbers SRR23047356-SRR23047361. The source data underlying the graphs for Fig. 2a, b, Fig. 4a–d, Fig. 5a–f and Fig. 7 in the paper is available in Supplementary Data 2.

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

## Acknowledgements

The study was financially supported by grants from the Scientific Research Foundation of the Third Institute of Oceanography, Ministry of Natural Resources (Grant number: 2019031), the Natural Science Foundation of Fujian Province, China (Grant number: 2020J01101), Xiamen Youth Innovation Foundation Project (Grant number: 3502Z20206100). We thank Professor Baobin Li at Fudan University, China, for the structural resolution of PET esterase (OQN32_06240) via AlphaFold2.

## Author contributions

W.B.G. designed the project. W.B.G. and Z.G.S. performed most of the experiments. X.Y. and J.J.D. performed some experiments on the enzyme characterization. W.B.G. and J.J.D. co-wrote the manuscript. W.B.G., J.J.D., and Z.Z.S. supervised experiment design and data analysis and provided funding.

## Competing interests

The authors declare no competing interests.
