## [Peer Review File · Communications Biology]

Reviewers' comments:

Reviewer #1 (Remarks to the Author):

This is a nicely driven work where authors isolate a marine PET biodegrading microorganism and further characterise a new PET hydrolase. It is great to see the overexpression work to identify this new enzyme, as well as the curious variation in activity/substrate-specificity depending on the pH. I have just some comments that need addressing:

INTRO

- Update references (e.g. global production numbers are from 2017; there are new estimates of plastics in the oceans; new studies on marine PET colonisation, etc).
- Delete unnecessary details (e.g. oyster and zooplankton feeding, the paper is not about ecotox; also, less on nutrient cycles and more of the current knowledge on PET biodegradation... there is actually nothing in the Intro on the current knowledge of PET biodeg).
- Massively reduce the last paragraph as results shouldn't be given here.

RESULTS

- Really!?!? Is this *Rhodococcus* motile??? This is a first! Please check.
- Fig. 2b is missing the control PET with no inoculum.
- Fig. 2c is missing a control of PET with no inoculum.
- Lines 188-193. It's not so 'obvious'. If cells were not induced by the presence of PET derivatives, the enzyme may not be produced and secreted. The cell fraction, in presence of PET derivatives can induce the production of the enzyme responsible of hydrolysis.
- During heterologous expression, how do authors know all 8 genes were correctly overexpressed? While it is very nice to see one successful (active) lysate, it is not possible to rule out that other genes may also code for a PET-like esterase. It may be these were not expressed correctly.
- While there is some indication that OQN32_06240 has some MHETase activity, it seems that *Rhodococcus* may have a 'more efficient' MHETase, right? Worth reflecting in fig 6.
- While authors nicely mention some genes are on a plasmid, please indicate the encoded location of all other genes (including the esterase). It would be interesting for them to all be on one same plasmid.
- Interesting transcriptomic data. But it would be nicer to see fold changes in table 1. Change.
- It's interesting to find this esterase in other Gram neg phyla. Could authors point out the transmembrane (or signal peptide) differences with *Rhodococcus*?

DISCUSSION

- While it is good to acknowledge old bibliography, there has been a large progress in recent years in this field (PET deg) which is NOT reflected. Please update.

M&Ms

- References are required. Authors need to give credit to those who develop techniques and methods (e.g. primers, software, etc etc). Check throughout.

Check language throughout the manuscript. There are parts that are even difficult to understand.

Reviewer #2 (Remarks to the Author):

COMMSBIO-23-1237-T

General remarks

In the manuscript titled " Biodegradation of PET plastic by a marine strain *Rhodococcus pyridinivorans* P23 with a membrane anchoring PET esterase in a biofilm model", Guo et al. isolated and characterized a new strain, *R. pyridinivorans* with depolymerization activity towards polyethylene terephthalate and identified and biochemically characterized PETase. Although well prepared, my concern is that the quality of the manuscript may not satisfy the standard of Communications Biology journal in the present form. It lacks novelty or highlighting the novelty aspects and necessary evidence to support some of the conclusions stated by the authors.

The authors present a compelling investigation of a novel membrane-bound PETase, test its activity on PET, BHET and MHET and prove the bacterium *R. pyridinivorans* P23 has the ability and does indeed utilize TPA. They continue with an interesting database search for organisms containing analogous PET-degradation capabilities. However, an extensive discussion and comparison of degradation rates to known PET-degrading systems both for the enzyme itself and the whole cells and/or biofilms is lacking.

Since the focus and novelty revolves around biofilm why were no standard biofilm-formation assays performed? Examining the wider substrates scope for the specific strain would provide valuable information, and determining the biofilm forming properties on different materials would also be important.

The number of figures should be reduced, and the existing figures and legends need to be clearer and presented alongside appropriate controls.

Introduction

Needs extensive reformatting and correction. Although lengthy, the information on PET-degrading microorganisms, PET-degrading esterases and PET waste handling is lacking. It only addresses the fate of PET in aquatic environments with no additional information on biofilm-forming strains in other environments and scientific/industrial methods for PET depolymerization.

Line 124 What exactly was evaluated?

Results

Line 133- 16S is not sufficient for species identification if the whole genome was used for phylogenetic analysis please indicate so.

Line 139-141- How can these appendages aid in PET colonization and what are they? The assumption that these are aggregations of PET degrading enzymes is unsubstantiated. Additionally, biofilm formation in liquid media by *R. pyridinivorans* P23 would be a much representation of a real-world scenario.

Line 158-163 - Can you please provide a quantitative measure of PET monomer release (ug/ml, uM) and how it relates to the weight loss?

Fig. 2-The intensity of the TPA signal barely changes during 7 days, how can this relate to the 4.28% weight loss? There is no control for potential PET-autohydrolysis presented in the graphs. Why is the OD600 and HPLC data presented for the course of 7 days and PET weight loss for 5 weeks? Where are these results for 5 weeks of incubation?

The y axis label in Fig. 2 b is missing and presenting the weight loss in mg and % is the same thing.

Line 176- BHET hydrolase activity in the inoculum? How was this proven?

Line 191- It is possible there are two enzymes, one acting on PET polymer (unknown location) and another one acting on BHET? Whole cell and recombinant experiments need to be clearly separated and discussed. In the present form, it is really difficult to make valid conclusions.

Line 211-214- These transmembrane esterases should be compared to known PET-degrading

enzymes.

Line 217- Why were the transmembrane domains removed? These domains could play an important role in correct protein folding and 3D structure.

Line 228-230- This statement has no point since the transmembrane domain was removed and all subsequent experiments were performed with the soluble form.

Fig. 3- B and C may be moved to supporting information

Line 268- How much enzyme was used?

Line 269- Some discussion on the fact most known PETases have an alkaline pH optimum should be included

Fig. 4- a, b, c, and d are not clear. Are these plots concerning the purified enzyme or whole-cell reactions?

Fig. 5- a, b, c, d, e, f, Why are the reaction times so different? This adds confusion to an already confusing figure. Purified PETase was incubated for 48 h and the cells for 30 min and 16 h, why? g, h, HPLC spectrum after what kind of degradation purified enzyme or cells?

Line 365-388 Were these experiments performed once? Can some statistical data be presented to corroborate these numbers?

Line 403-437- It should be made clear that this is a proposed degradation mechanism and not all of the claims such as trapping of PET monomers has been experimentally confirmed.

Line 455-479- Are these bacteria capable of forming biofilm and do they reside in the same ecological niches as *R. pyridinovorans* P23 for them to potentially have the same degradation capabilities? Do all of these esterases possess a transmembrane domain since the identity hits of 60% can easily exclude the TM domain.

Additionally, how does this fit into a broader group of organisms which contain PETases (not only OQN32_06240) and the TPA degradation pathway.

Discussion

Line 515-537- This section is basically "Biodegradation of PET by marine strains attached to plastic surface forming biofilm" from the Results just rewritten. This section should be removed from the Results section and be written in its entirety in the Discussion.

Line 554-556- How does the degradation rate of *R. pyridinovorans* P23 compare to other whole-cell PET degradation systems both natural and synthetic since a degradation rate of 4% over the course of 5 weeks doesn't seem viable for large scale application.

Materials and Methods

Line 584- mineral salt medium contains 2.5 g/L of carbamide (urea). Although uncommon carbamide could be used as a carbon source (DOI: 10.3389/fmicb.2019.01064)

Line 586- Marine 2216 agar composition is missing.

Line 591- The 16S rRNA gene can only be used to identify the genus of the strain

Line 675- How long was the induction?

Supplementary material

Supplementary figure 2- Needs additional information in the description.

Supplementary Figure 3- The y axis needs to be labeled and an additional axis is missing? The concentrations of BHET, MHET and TPA are presented in % in the description and in mg/ml in the graphs.

Supplementary Figure 10- This part is missing in the Materials and Methods section which concentrations were used, "When TPA/MHET solution reached a certain pH, the concentration of TPA/MHET was determined by HPLC" what concentration....The concentrations in the figure differ so do the pH. This figure is very confusing and unclear.

Reviewer #3 (Remarks to the Author):

Comments for COMMSBIO-23-1237-T

English is well written and logical. However, the size of the manuscript is too big, which looks a little redundant. If possible, concise the overall description, which gives stronger impression to readers. The manuscript deserves for publication, but still it includes some points to be revised.

Introduction

Line 82: It should be smaller molecules rather than smaller monomers.

Results

Fig. 1: Indicate a scale bar in photographs. Either c or d is enough.

Table S1: Define TM. Instead of outside, "predicted mature protein" is better.

In addition, show the location of each gene in the total genome (chromosomal or plasmid-located).

Lines 334-335: Not only Kcat/Km value for MHET, but also that for BHET are to be described.

Lines 354-362: Genes for TPADO, DCDDH and others are from *I. sakaiensis* (Table S2). Instead of them the corresponding genes in your strain must be indicated, showing that they correspond to those in *I. sakaiensis*. Is PCA defined in the text?

Line 379: Indicate the corresponding figure or table after As we mentioned above. It is so difficult to confirm the previous description or data.

Fig. 6a is not shown in the text. Show the corresponding part in the text.

Discussion

From the degradation rate of PET (you described 4.03 mg weight loss from the original weight 92.9-96.2 mg at 30 °C for 5 weeks), it is not practical to apply your strain, even for PET existing in marine environments (lower than 30 °C). The point of this paper is that PET degrading microorganisms are born by the existence of PET. They might contribute to the disappearance of PET taking long time, although its practical contribution to clean up the ecosystem is suspicious as the degradation rate is too low, compared to the amount of PET waste released. I never want to deny your efforts and values of this paper, but you should describe the realistic phase of validity of bioremediation, using your own and other PET utilizing microorganisms. If they work efficiently enough for bioremediation, they will cause deterioration of PET products, resulting in damaging our society, because it is so difficult to prevent microorganisms from their invasion into our lives. Durability of products is requisite even for plastics. The value of this paper is thought to be different from application for bioremediation and biorecycling. I am impressed by that microorganisms can adapt PET at low temperature and that the novel PET hydrolase was found. The original paper of IsPETase insisted that it could be applied for biorecycling, instead of thermophilic PET hydrolases, but the idea has already been denied by several review papers; the most recent one is Chem. Rev. An article for ASAP (Review); Publication date (Web; March 14, 2023) DOI: 10.1021/acs.chemrev.2c00644.

Materials and methods

Line 581: Specify PET powder (origin, and preparation to make it etc.).

**Key Laboratory of Marine Biogenetic Resources,
Third Institute of Oceanography,
Ministry of Natural Resources,
184 Daxue Road, Xiamen 361005, Fujian, China.**

Aug. 29, 2023

Dear reviewers,

Thanks very much for your work on our manuscript. It is great honor for us to receive the comments, which give us lots of valuable elicitation and guidance. The following table shows our answers to the relative comments and critiques. A revised manuscript with marked track changes would be appended. All the modifications were done according to your comments. Although we have made every effort to revise the manuscript, it is inevitable that there are still some shortcomings. I hope the reviewers can point out these weak points if have, that we are willing to make revisions based on everyone's opinions further, so that our findings can be published.

With best regards!

Sincerely Yours,

Dr. Wenbin Guo

Key Laboratory of Marine Biogenetic Resources,

Third Institute of Oceanography, Ministry of Natural Resources,

Xiamen 361005, Fujian, China.

Reviewers' comments/critiques	Answers and changes
Referee #1: This is a nicely driven work where authors isolate a marine PET biodegrading microorganism and further characterise a new PET hydrolase. It is great to see the overexpression work to identify this new enzyme, as well as the curious variation in activity/substrate-specificity depending on the pH. I have just some comments that need addressing:	Thank you for your positive comments.
INTRO - Update references (e.g. global production numbers are from 2017; there are new estimates of plastics in the oceans; new studies on marine PET colonisation, etc).	Reference 2 and 8 had been updated. The related data had also been updated to 2019 and thereafter. Please see line 48- 61.
- Delete unnecessary details (e.g. oyster and zooplankton feeding, the paper is not about ecotox; also, less on nutrient cycles and more of the current knowledge on PET biodegradation... there is actually nothing in the Intro on the current knowledge of PET biodeg).	The description about oyster and zooplankton and their related ref 9 and 10 were deleted. Please see line 63. A detailed description of PET biodegradation by microorganism and enzymes were added in the introduction. Please see line 87- 114.
- Massively reduce the last paragraph as results shouldn't be	The last paragraph was massively reduced.

given here.	
RESULTS - Really!?!? Is this Rhodococcus motile??? This is a first! Please check.	Sorry for the low-level error. And it was revised as non-motile. Line 160.
- Fig. 2b is missing the control PET with no inoculum.	The negative control of PET weight loss in cultivation without inoculum was added to Fig. 2b.
- Fig. 2c is missing a control of PET with no inoculum.	The HPLC spectrum of 7d PET film cultivation without inoculum was added to Fig. 2c.
- Lines 188-193. It's no so 'obvious'. If cells were not induced by the presence of PET derivatives, the enzyme may not be produced and secreted. The cell fraction, in presence of PET derivatives can induce the production of the enzyme responsible of hydrolysis.	In Fig. S4, MHET accumulation by cells is obvious higher than that by supernatant. In the BHET hydrolysis reaction solution, the enzyme is no possible to be produced after induction of BHET in the phosphate buffer by the cells as the limited nutrient substance.
- During heterologous expression, how do authors know all 8 genes were correctly overexpressed? While it is very nice to see one successful (active) lysate, it is not possible to rule out that other genes may also code for a PET-like esterase. It may be these were not expressed correctly.	We used SDS-PAGE to detect the heterologous expression level of all 8 genes in E. coli BL21(DE3). The SDS-PAGE electropherogram was provided in Fig. S7 and showed that all the 8 genes were successfully and correctly expressed in E. coli.
- While there is some indication that OQN32_06240 has some MHETase activity, it seems that Rhodococcus may have a 'more efficient' MHETase, right? Worth reflecting in fig 6.	As the purified PET esterase (OQN32_06240) could degrade MHET in acid environment which was in accordance with the whole-cell biocatalyst towards MHET. This indicates that there is a high possibility the enzyme displayed on the surface of the cells is PET esterase (OQN32_06240). When cells of R. pyridinivorans P23 were used as whole-cell biocatalyst (OD₆₀₀ of 1.0) towards 1.0 mM BHET at 30 °C in pH 4.5 buffer, 34.5% BHET was transformed into MHET in 30 minutes (Fig. 5b). When the reactions were performed for 5 h using 1.0 mM MHET as substrate for whole-cell biocatalyst (OD₆₀₀ of 1.0), 20.6% of MHET was transformed into TPA in pH 4.5 (Fig. 5e). Actually, the enzyme t is also powerful towards MHET in acid conditions. Cultivation of R. pyridinivorans P23 using MHET as sole carbon and energy source showed that it grown slowly in accordance with the MHETase activity of PET esterase (OQN32_06240) in pH 7 condition (Fig. S3b). Sequence searching of Is-MHETase like enzyme in the whole genome of R. pyridinivorans P23 showed that no significant hit was obtained. Therefore, it is not possible that there are other efficient MHETases.
- While authors nicely mention some genes are on a plasmid, please indicate the encoded location of all other genes (including the esterase). It would be interesting for them to all be on one same plasmid.	The locations of PET esterase encoding gene and genes involving in the PET degradation pathway were all indicated in the manuscript. Only the TPA degradation operon (OQN32_25270- OQN32_25290)

	locates on the pA plasmid. See line 322-334.
- Interesting transcriptomic data. But it would be nicer to see fold changes in table 1. Change.	The fold change of transcriptional data of each gene in Table 1 was provided accordingly.
- It's interesting to find this esterase in other Gram neg phyla. Could authors point out the transmembrane (or signal peptide) differences with Rhodococcus?	The Annotree server was utilized to search the coexistence of PET esterase (PF07859, K01066) and large subunit of the oxygenase component of TPADO (PF00848, K16319) homologs in bacteria. The genome hits indicate that the coexistence of two enzymes with pfam number of PF07859 and PF00848, or with ko number of K01066 and K16319. However, it is not clear whether the corresponding enzymes in those hit genome has the transmembrane domain or not.
DISCUSSION - While it is good to acknowledge old bibliography, there has been a large progress in recent years in this field (PET deg) which in NOT reflected. Please update.	The references were updated with Ref. 44 and 45.
M&Ms - References are required. Authors need to give credit to those who develop techniques and methods (e.g. primers, software, etc etc). Check throughout.	Actually, the 16S rDNA sequence was obtained from whole genome sequencing as it is long enough to construct the phylogenetic tree. The primers were deleted from M&Ms. The reference about the Annotree server was cited. See Ref 57.
Check language throughout the manuscript. There are parts that are even difficult to understand.	We checked the language throughout the manuscript and tried our best to modify some inappropriate language expressions as much as possible.
Referee #2: In the manuscript titled " Biodegradation of PET plastic by a marine strain Rhodococcus pyridinivorans P23 with a membrane anchoring PET esterase in a biofilm model", Guo et al. isolated and characterized a new strain, R. pyridinivorans with depolymerization activity towards polyethylene terephthalate and identified and biochemically characterized PETase. Although well prepared , my concern is that the quality of the manuscript may not satisfy the standard of Communications Biology journal in the present form. It lacks novelty or highlighting the novelty aspects and necessary evidence to support some of the conclusions stated by the authors. The authors present a compelling investigation of a novel membrane-bound PETase, test its activity on PET, BHET and MHET and prove the bacterium R. pyridinivorans P23 has the ability and does indeed utilize TPA. They continue with an interesting database search for organisms containing analogous PET-degradation capabilities. However, an extensive discussion and comparison of degradation rates to known PET-degrading systems both for the enzyme itself and the whole cells and/or	An extensive discussion and comparison of degradation rates to known PET-degrading systems both for the enzyme itself and the cells of R. pyridinivorans P23 was provided. See line 469-477 and line 483- 495. In this study, we aimed to explore the PET hydrolase resource and shed light on its characteristic on PET, BHET and MHET degradation. Fortunately, we found out that PET esterase (OQN32_06240) not only had the ability to degrade PET and BHET, but MHET in acid conditions. As the strain R. pyridinivorans P23 was isolated from the marine environment with alkaline sea water. Therefore, from these results, we proposed a biofilm model on PET plastic for R. pyridinivorans P23 to explain how this strain survived in the marine environment. See line 446- 461. As the reviewer pointed out that standard biofilm-formation assay was not performed, we moved the proposed biofilm degradation model from "results" to the "discussion" section. Some figures were moved to the supplementary

biofilms is lacking. Since the focus and novelty revolves around biofilm why were no standard biofilm-formation assays performed? Examining the wider substrates scope for the specific strain would provide valuable information, and determining the biofilm forming properties on different materials would also be important. The number of figures should be reduced, and the existing figures and legends need to be clearer and presented alongside appropriate controls.	materials such as Fig. 3b and 3c. Some figure legends were revised to be clearer, and some negative controls were added in the figures such as Fig. 2b and Fig. 2c.
Introduction Needs extensive reformatting and correction. Although lengthy, the information on PET-degrading microorganisms, PET-degrading esterases and PET waste handling is lacking. It only addresses the fate of PET in aquatic environments with no additional information on biofilm-forming strains in other environments and scientific/industrial methods for PET depolymerization.	A detail description of PET biodegradation by microorganism and enzymes were added in the introduction. Please see line 87- 114.
Line 124 What exactly was evaluated?	The microorganisms with similar PET-degradation capabilities were evaluated. Based on the comments of other reviewers, this paragraph was massively reduced. And this sentence was deleted.
Results Line 133- 16S is not sufficient for species identification if the whole genome was used for phylogenetic analysis please indicate so.	Analysis of the 16S rRNA gene sequence in EzbioCloud database (https://www.ezbiocloud.net/) showed that it belonged to the Rhodococcus genus and exhibited the highest 16S rRNA gene sequence similarity with type strain Rhodococcus pyridinivorans DSM 44555(T) (99.93%) with only 1 base variation in 1444 bases. Construction of phylogenetic tree with type strain Rhodococcus pyridinivorans DSM 44555(T) and other type strains of Rhodococcus genus showed that it was most closely related to Rhodococcus pyridinivorans DSM 44555(T). See Fig. S2. As there are 6 type strains with 16S sequences similarity of more than 98% to strain P23 (figure below). The possibility of strain P23 being a new species is quite low. Based on the analysis, strain P23 was identified as Rhodococcus pyridinivorans. Average Nucleotide Identity (ANI) value of 95.16% was obtained via ANI Calculator in EzbioCloud using strain P23 and R. pyridinivorans DSM44555(T) (GCA_900105195.1) genomes. Line 152- 159.

List of hits from EzBioCloud 16S database

Select hits by database

Tasks	Hit taxon name	Hit strain name	Accession	Similarity	Variation ratio	
[ ]	Rhodococcus pyridinivorans	DSM 44555(T)	LRRI01000001	99.93	1/1444	Bacteria:Ac
[ ]	Rhodococcus biphenylivorans	TGW(T)	KJ545454	99.92	7/1444	Bacteria:Ac
[ ]	Rhodococcus gordoniae	DSM 44699(T)	LPZNO1000053	99.24	11/1444	Bacteria:Ac
[ ]	Rhodococcus rhodochrous	NBRC 16099(T)	BBXP01000056	98.82	17/1444	Bacteria:Ac
[ ]	Rhodococcus lactis	DW1519(T)	KP942300	98.83	21/1425	Bacteria:Ac
[ ]	Rhodococcus artemisiae	YIM 65754(T)	GU367155	98.13	27/1441	Bacteria:Ac
[ ]	Rhodococcus coprophilus	NBRC 100603(T)	BDAM01000020	97.71	33/1444	Bacteria:Ac
[ ]	Rhodococcus ruber	DSM 43398(T)	LRRL01000064	97.58	35/1444	Bacteria:Ac
[ ]	Rhodococcus electrodiphilus	JC435(T)	LT630357	97.51	35/1406	Bacteria:Ac
[ ]	Rhodococcus phenolicus	DSM 44812(T)	LRRH01000094	97.44	37/1444	Bacteria:Ac

Line 139-141- How can these appendages aid in PET colonization and what are they? The assumption that these are aggregations of PET degrading enzymes is unsubstantiated. Additionally, biofilm formation in liquid media by *R. pyridinivorans* P23 would be a much representation of a real-world scenario.

At present, the real functionality of appendages is unknown. And how can they aid in PET colonization needs to be revealed in the future. Accordingly, the assumption that the appendages work in aggregation of PET-degrading enzymes was deleted. Line 165.

Line 158-163 - Can you please provide a quantitative measure of PET monomer release (ug/ml, uM) and how it relates to the weight loss?

From the HPLC chromatogram, BHET, MHET and TPA are the three monomeric product released from PET degraded by *R. pyridinivorans* P23. BHET could be transformed into MHET which could be transformed into TPA further. TPA was served as the substrate of *R. pyridinivorans* P23 and would be transported into cell for utilization. As a result, these three monomeric products were in dynamic concentrations. More importantly, TPA was the final product of PET degradation and would be utilized. These three monomeric products measured only representing their current state in the cultivation supernatant. Therefore, the totally released amount of these three monomeric products could not be quantitatively measured to relate to the weight loss of PET.

Fig. 2-The intensity of the TPA signal barely changes during 7 days, how can this relate to the 4.28% weight loss? There is no control for potential PET-autohydrolysis presented in the graphs.

The TPA intensity representing its current state in the cultivation supernatant. TPA could be obtained from PET biodegradation and then utilized by cells. Therefore, there is no quantitative correlation in HPLC determined TPA with the 4.28% weight loss of PET. The HPLC chromatogram of negative control without inoculation was added in the Fig. 2c.

Why is the OD600 and HPLC data presented for the course of 7 days and PET weight loss for 5 weeks? Where are these results for 5 weeks of incubation?

As the biodegradation of PET by *R. pyridinivorans* P23 was relatively slow, the cultivation process of *R. pyridinivorans* P23 with PET as sole carbon source needs several days (7 days) to make sure the weight loss of PET plastic be significantly different to the control. Meanwhile, the cell growth of *R. pyridinivorans* P23 in PET cultivation turn slowly after 7 days. So, mineral medium needs to be replaced with fresh one after 7 days cultivation. In the 5 weeks cultivation, it's a 5 repeat of 7 days cultivation except

	that the PET was obtained from last round of 7 days cultivation. This would prove that PET plastic could be continuously degraded by R. pyridinivorans P23 if the cultivation conditions are suitable. The cultivations take long time. It must be ensure that the PET plastic used for weighing after cultivation is intact in form. Due to the effect of mechanical shear force, longer cultivation time means an increased risk of plastic film damage in formation. There, the whole cultivation time was 5×7 days accordingly.
The y axis label in Fig. 2 b is missing and presenting the weight loss in mg and % is the same thing.	The yaxis label in Fig. 2b was added.
Line 176- BHET hydrolase activity in the inoculum? How was this proven?	In the BHET cultivation, BHET was degraded into MHET rapidly as soon as the inoculation of R. pyridinivorans P23 cells which was prepared from 2216 marine broth cultivation in a test tube (Fig. S3a). If the BHET hydrolase was not in the inoculum, there are not enough time for the enzyme synthesis and secretion. In subsequent experiments, BHET hydrolysis by cells collected from 2216 marine broth cultivation proved that the BHET hydrolase activity was in the cells but not supernatant (Fig. S4).
Line 191- It is possible there are two enzymes, one acting on PET polymer (unknown location) and another one acting on BHET? Whole cell and recombinant experiments need to be clearly separated and discussed. In the present form, it is really difficult to make valid conclusions.	According to our knowledge, many reported PET degrading enzymes exhibit BHET hydrolase activity simultaneously. These PET degrading enzymes includes Is-PETase, TtH, FsC, HiC and BsEstB (Yoshida S. et al, Science, 2016, Table S2). In this study, PET esterase (OQN32_06240) was heterologously expressed, purified and proved to degrade PET, BHET and MHET (MHET only in acid pH). Meanwhile, the activity profiles of purified PET esterase (OQN32_06240) and whole-cell biocatalyst of R. pyridinivorans P23 towards PET, BHET and MHET in different pH conditions are the same. The legends of Fig. 4 and 5 were revised more clearly to separate purified enzyme and the cell biocatalyst.
Line 211-214- These transmembrane esterases should be compared to known PET-degrading enzymes.	These 8 potential PET hydrolases were compared to known PET degrading enzymes Is-PETase, LCC and TtH with a phylogenetic tree constructed using Mega 6.0 (Fig. S6).
Line 217- Why were the transmembrane domains removed? These domains could play an important role in correct protein folding and 3D structure.	These transmembrane domains exist at the N-terminal of the enzyme, while the C-terminal domain is the main catalytic domain according to sequence analysis of these enzymes. In order to make sure proteins in soluble expression in E. coli, the usual approach is removing the hydrophobic transmembrane domain in the proteins. However, the concerns of the reviewers also exist, fortunately we found the correct enzyme

	and expressed the active functional enzyme in the end.
Line 228-230- This statement has no point since the transmembrane domain was removed and all subsequent experiments were performed with the soluble form.	Analysis of transmembrane domain can help us understand the anchoring state of this enzyme on the surface of cell membrane.
Fig. 3- B and C may be moved to supporting information	Fig. 3b and c were both moved to supplementary materials.
Line 268- How much enzyme was used?	The final enzyme concentration in the reaction solution is 1.0 μ M.
Line 269- Some discussion on the fact most know PETases have an alkaline pH optimum should be included	Revised accordingly.
Fig. 4- a, b, c, and d are not clear. Are these plots concerning the purified enzyme of whole-cell reactions?	Fig. 4a, c and d are concerning the purified enzyme from heterologous expression of PET esterase (OQN32_06240) in E. coli . Fig. 4b is concerning the whole-cell reactions of R. pyridinivorans P23. The Figure captions were modified to be clearer, more accurate, and easier to understand.
Fig. 5- a, b, c, d, e, f, Why are the reaction times so different? This adds confusion to an already confusing figure. Purified PETase was incubated for 48 h and the cells for 30 min and 16 h, why? g, h, HPCL spectrum after what kind of degradation purified enzyme or cells?	BHET could be hydrolyzed into MHET by purified PET esterase (OQN32_06240) or cells of R. pyridinivorans P23. Meanwhile, MHET was served as the substrate of the same enzyme in acid conditions. In order to observe the dynamic transformation of BHET into MHET (Fig. 5b) and then TPA (Fig. 5c), 30 min and 16 h of reaction time were applied in the reactions. Results showed that the reaction time were proper. g, HPLC spectrum of BHET degradation by purified PET esterase (OQN32_06240) and R. pyridinivorans P23 cells in pH 4.5. h, HPLC spectrum of MHET degradation by purified PET esterase (OQN32_06240) and R. pyridinivorans P23 cells in pH 4.5. The figure captions were revised.
Line 365-388 Were these experiments performed once? Can some statistical data be presented to corroborate these numbers?	Two biological replicates each were used in the preparation of RNA samples from the three different substrate culture conditions. The RPKM values in Table 1 were revised as mean \pm standard deviation. Meanwhile, the logarithmic values of Fold Change of RPKM values were provided.
Line 403-437- It should be made clear that this is a proposed degradation mechanism and not all of the claims such as trapping of PET monomers has been experimentally confirmed.	This section was moved to the “Discussion” section with the section title of “Proposed PET biodegradation by marine strains attached to plastic surface forming biofilm”. Line 388-422.
Line 455-479- Are these bacteria capable of forming biofilm and do they reside in the same ecological niches as R. pyridinivorans P23 for them to potentially have the same degradation capabilities? Do all of these esterases possess a transmembrane domain since the identity hits of 60% can easily exclude the TM domain.	It is not clear whether these bacteria can form biofilms or not. It is also not clear that what ecological niches these bacteria reside. The purpose of this paragraph is to inform readers that there are microbial communities in the natural environment similar to R. pyridinivorans P23 that can degrade and

Additionally, how does this fit into a broader group of organisms which contain PETases (not only OQN32_06240) and the TPA degradation pathway.	utilize PET plastics. The transmembrane domains of these esterases are also unclear. In searching microbial groups with other PETase, it has first to determine the pfam number and ko number of the enzyme.
Discussion Line 515-537- This section is basically “Biodegradation of PET by marine strains attached to plastic surface forming biofilm” from the Results just rewritten. This section should be removed from the Results section and be written in it’s entirety in the Discussion.	As the corresponding section of “results” was moved to “Discussion”, line 515-537 in was removed from “Discussion”.
Line 554-556- How does the degradation rate of R. pyridinivorans P23 compare to other whole-cell PET degradation systems both natural and synthetic since a degradation rate of 4% over the course of 5 weeks doesn’t seem viable for large scale application.	We compared the degradation rate of R. pyridinivorans P23 to other degradation systems. Although the efficiency of R. pyridinivorans P23 in degrading PET plastics currently does not reach the level of industrial application, as it was discovered in the marine environment which is the gathering place of discarded plastic waste in the environment, this bacterium and its similar PET degradation and assimilation functional microbial groups will play an important roles in the bioremediation of marine plastic pollution. Line 490-495.
Materials and Methods Line 584- mineral salt medium contains 2.5 g/L of carbamide (urea). Although uncommon carbamide could be used as a carbon source (DOI: 10.3389/fmicb.2019.01064)	Usually, urea is only used as a nitrogen source. In our cultivation experiments, cultivation of R. pyridinivorans P23 in mineral salt medium without addition of PET was used as negative control. No growth of R. pyridinivorans P23 in the negative control was observed (Fig. 2a).
Line 586- Marine 2216 agar composition is missing.	The 2216 marine agar medium was purchased from Haibo corporation, Qingdao, China. The composition of 2216 marine agar was provided in the revised manuscript. Line 520- 525.
Line 591- The 16S rRNA gene can only be used to identify the genus of the strain	The 16S rRNA gene can be used to identify the species through EzbioCloud database (https://www.ezbiocloud.net/) which contains type strains. Furthermore, a phylogenetic tree was constructed using 16S rRNA genes of strain P23 and other Rhodococcus species type strains showing that strain P23 was more related to type strain Rhodococcus pyridinivorans DSM44555(T). Average Nucleotide Identity (ANI) value of 95.16% was obtained via ANI Calculator in EzbioCloud using strain P23 and R. pyridinivorans DSM44555(T) (GCA_900105195.1) genomes. Based on these results, we tend to believe that P23 belongs to R. pyridinivorans. In Fig. S2, type strains are denoted via T.
Line 675- How long was the induction?	16 hours.

Supplementary material Supplementary figure 2- Needs additional information in the description.	More information was provided in the description of Fig. S2. Type strains are denoted via T in Fig. S2.
Supplementary Figure 3- The y axis needs to be labeled and a additional axis is missing? The concentrations of BHET, MHET and TPA are presented in % in the description and in mg/ml in the graphs.	The y axis in Fig. S3 were labeled. The description was revised as mg/ml or g/l.
Supplementary Figure 10- This part is missing in the Materials and Methods section which concentrations were used, “When TPA/MHET solution reached a certain pH, the concentration of TPA/MHET was determined by HPLC” what concentration....The concentrations in the figure differ so do the pH. This figure is very confusing and unclear.	The original purpose of this experiment was to tell people that the carboxyl groups of MHET and TPA can make pure water and seawater acidic, but it can be easily understood without experimental data. In order to avoid misunderstandings, we have decided to delete this supplementary image.
Referee #3: English is well written and logical. However, the size of the manuscript is too big, which looks a little redundant. If possible, concise the overall description, which gives stronger impression to readers. The manuscript deserves for publication, but still it includes some points to be revised.	Thank you for your positive comments. Fig. 3b and Fig. 3c were moved to supplementary materials to make the manuscript concise.
Introduction Line 82: It should be smaller molecules rather than smaller monomers.	Revised accordingly.
Results Fig. 1: Indicate a scale bar in photographs. Either c or d is enough.	A 3 μ m scale bar has already been provided in the lower right corner of corresponding image. Fig. 1d was removed.
Table S1: Define TM. Instead of outside, “predicted mature protein” is better. In addition, show the location of each gene in the total genome (chromosomal or plasmid-located).	Revised accordingly in Table S1.
Lines 334-335: Not only Kcat/Km value for MHET, but also that for BHET are to be described.	Kcat/Km value for BHET had already been provided in the section of “Characterization of PET esterase (OQN32_06240) towards BHET”. Line 296- 298.
Lines 354-362: Genes for TPADO, DCDDH and others are from I. sakaiensis (Table S2). Instead of them the corresponding genes in your strain must be indicated, showing that they correspond to those in I. sakaiensis . Is PCA defined in the text?	These genes/enzymes are from R. pyridinivorans P23. The ORF# in Table S2 was revised more clearly to understand. The full name of PCA was provided in line 329 in the revised manuscript.
Line 379: Indicate the corresponding figure or table after As we mentioned above. It is so difficult to confirm the previous description or data.	Revised accordingly.
Fig. 6a is not shown in the text. Show the corresponding part in the text.	The corresponding part of Fig. 6a in the text was provided in the section of “PET degradation pathway in R. pyridinivorans P23”. See line 321.
Discussion From the degradation rate of PET (you described 4.03 mg weight loss from the original weight 92.9-96.2 mg at 30 C for 5 weeks), it is not practical to apply your strain, even for PET existing in marine environments (lower than 30 C). The point of	Thank you for your suggestions. The oceanic environment has been recognized as a major sink for plastic. This has brought many impacts and hazards to marine ecology and the environment. The prerequisite for correctly evaluating these impacts is to understand

this paper is that PET degrading microorganisms are born by the existence of PET. They might contribute to the disappearance of PET taking long time, although its practical contribution to clean up the ecosystem is suspicious as the degradation rate is too low, compared to the amount of PET waste released. I never want to deny your efforts and values of this paper, but you should describe the realistic phase of validity of bioremediation, using your own and other PET utilizing microorganisms. If they work efficiently enough for bioremediation, they will cause deterioration of PET products, resulting in damaging our society, because it is so difficult to prevent microorganisms from their invasion into our lives. Durability of products is requisite even for plastics. The value of this paper is thought to be different from application for bioremediation and biorecycling. I am impressed by that microorganisms can adapt PET at low temperature and that the novel PET hydrolase was found. The original paper of IsPETase insisted that it could be applied for biorecycling, instead of thermophilic PET hydrolases, but the idea has already been denied by several review papers; the most recent one is Chem. Rev. An article for ASAP (Review); Publication date (Web; March 14, 2023) DOI: 10.1021/acs.chemrev.2c00644.	the interaction between marine microorganisms and plastics. Specifically, regarding PET plastics, in addition to the microorganisms described in this paper that degrade PET and utilize TPA, we also studied the microbial groups that degrade PET and utilize EG in the marine environment. We will describe this in a separate paper. Ninety percent of the sea water in the ocean is below 5 °C throughout the year (Methe et al. 2005). The average temperature of the ocean is very low, and the process of enzymatic degradation of plastics is very slow under such temperature conditions. However, this also highlights the importance of the large community of PET plastic degrading bacteria in the ocean. However, currently, the understanding of the microorganisms and enzymes that degrade PET plastics in the ocean is not sufficient, and their understanding of the degradation process of PET plastics is also insufficient. This is also the significance of this paper. We discuss the potential of these microorganisms in the bioremediation in the marine environment. See line 481-495.
Materials and methods Line 581: Specify PET powder (origin, and preparation to make it etc.).	PET powder was crushed from granular PET purchased from Sigma-Aldrich, USA with lot number of 429252. See line 635.

Reviewers' comments:

Reviewer #1 (Remarks to the Author):

Authors have addressed my comments

Reviewer #2 (Remarks to the Author):

The manuscript has been extensively modified in accordance with reviewer's comments. There are still some issues that can be dealt with during the editorial process (different font and style of References section, units not all in SI format). I give positive opinion regarding publication of this manuscript.

Reviewer #3 (Remarks to the Author):

Comments for Revision for COMMSBIO-23-1237A

English writing is poor, which looks without having English edition by native speakers (Native speakers are not always sufficient for scientific papers. Professional editors could check not only spelling and grammar, but also the logic and composition of the paper.). Many grammatical errors are found throughout the paper and sentences are often redundant. For example, any native speaker misses "detailed characterized (Line 142)". Lines 142-144 can be replaceable with "The PET esterase was characterized to show its relevance to the PET degradation by *R. pyridinivorans* P23". The revised paper is shorter than the original one. However, still description is redundant, and many sentences can be concisely described without losing their meanings.

Introduction

1. Line 100: Instead of Ref. 24, cite the newer one(s) including various microbial origins. Microorganisms used in Ref. 24 is no more sufficient. Until 2016, more microorganisms had already been documented.
2. Lines 103-the end of the paragraph: The description misleads readers, especially those unfamiliar with this area. Revise totally. It has already been denied that the wild-type Is-PETase has better activity than PET hydrolases known up to 2016. Ref 30 did not try to improve Is-PETase. Even now, no mutants derived from Is-PETase surpassed thermophilic PET hydrolases. Is-PETase (Yoshida et al.) insisted that the enzyme is better at 30°C than thermophilic enzymes (unfair description) and even 30°C is higher than the temperature in marine environments. Ref. 31 is not based on the sequence of Is-PETase (based on the amino acid sequences based on various PET hydrolases, as the basic structure of Is-PETase is homologous to those of thermophilic enzymes). This part must be rewritten, for example, as follows: PET hydrolases are categorized into marine and terrestrial ones (Danson et al.). The major microorganisms in marine environment are --- and those in terrestrial environment are---. As PET hydrolases derived from mesophilic microorganisms, SM14est -----.
3. Line 142: Grammatically wrong with "detailed characterized".

Results

1. Lines 253-254: distinct from MHETase and Abhydrolase_5, but homologous to abhydrolase_3 including IsPETase

2. Line 257: BHET, not monomeric BHET

3. Line 261: "monomeric products" are strange. Why are they monomeric? Monomeric is used, for example, like that MHET is a monomeric unit for PET. No TPA and BHET polymers exist. Therefore, they cannot be monomers for their polymers. Authors confuse monomeric materials and small molecules. Revise the paper totally because erroneous words are used throughout the paper.

4. Lines 269 and 275: relatively high, not relative high. English edition by a professional native speaker (available) is requisite. There are too many grammatical errors.

5. Line 275: relative high enzymatic?

6. Lines 276-280 : ? 7. Lines 286-287: Probably with E. coli. Then, suggest this. 8. PET plastic is strange. PET is enough. We never express PE, PVC and any plastics like PE plastic, PVC plastic etc.

Discussion

1. This section is categorized in subsection, but I do not think that subsections are appropriate in Discussion. The description of the first subsection, especially lines 390-413, is not discussion but results (should be included in Results).

2. Line 452: Is-PETase (WT) has no H|MHET hydrolase activity.

3. Line 455: ---pH 8, which is not the optimum pH to activate MHET hydrolase---- Optimum pH cannot be used for activation.

4. Line 462: The section title is inappropriate.

5. Line 464: As described above, PET esterase is close to Abhydrolase_3 including Is-PETase.

6. Lines 472-475: Rewrite the sentence. Show the reference for Ple628 and PE-H, separately and respectively.

7. Lines 491-495: Rewrite. How do you think to apply your strain for the bioremediation? The paper suggests the slow decay of PET in marine environments,

Materials and Methods

1. Line 521: The 2216 marine broth is better, as figures at the top of a sentence must be written in alphabet like twenty-two hundreds and sixteen

**Key Laboratory of Marine Biogenetic Resources,
Third Institute of Oceanography,
Ministry of Natural Resources,
184 Daxue Road, Xiamen 361005, Fujian, China.**

Sept. 21, 2023

Dear reviewers,

Thanks very much for your earnest work on our manuscript. We are delighted to have the opportunity to make revisions to the manuscript. It is great honor for us to receive the comments, which give us lots of valuable elicitation and guidance. The following table shows our answers to the relative comments and critiques. A revised manuscript with marked changes would be appended. Although we have made every effort to revise the manuscript, it is inevitable that there are still some shortcomings. I hope the reviewers can point out these weak points if have, that we are willing to make revisions based on everyone's opinions further, so that our findings can be published.

With best regards!

Sincerely Yours,

Dr. Wenbin Guo

Key Laboratory of Marine Biogenetic Resources,

Third Institute of Oceanography, Ministry of Natural Resources,

Xiamen 361005, Fujian, China.

Reviewers' comments/critiques	Answers and changes
Reviewer #1 (Remarks to the Author): Authors have addressed my comments	Thank you for your positive comments.
Reviewer #2 (Remarks to the Author): The manuscript has been extensively modified in accordance with reviewer's comments. There are still some issues that can be dealt with during the editorial process (different font and style of References section, units not all in SI format). I give positive opinion regarding publication of this manuscript.	Thank you for your positive comments.
Reviewer #3 (Remarks to the Author): Comments for Revision for COMMSBIO-23-1237A English writing is poor, which looks without having English edition by native speakers (Native speakers are not always sufficient for scientific papers. Professional editors could check not only spelling and grammar, but also the logic and composition of the paper.). Many grammatical errors are found throughout the paper and sentences are often redundant. For example, any native speaker misses “detailed	Revised accordingly in line 140- 142. English writing was checked through the whole manuscript, and some spelling and grammar mistakes were revised. For instance, “product” in line 174 was revised as “products”. Line 198, “locates” was deleted. Line 229, lysates were Line 267, of Line 270, solutions

characterized (Line 142)”. Lines 142-144 can be replaceable with “The PET esterase was characterized to show its relevance to the PET degradation by R. pyridinivorans P23”. The revised paper is shorter than the original one. However, still description is redundant, and many sentences can be concisely described without losing their meanings.	Line 282, 286, 291, 292, buffers Line 284, hours Line 310, “the” removed Line 313, conditions Line 359, delete “The” Line 360, might Line 362, genome Line 365, 374, of Rewrite sentence in line 482- 484. Line 489, role
Introduction 1. Line 100: Instead of Ref. 24, cite the newer one(s) including various microbial origins. Microorganisms used in Ref. 24 is no more sufficient. Until 2016, more microorganisms had already been documented.	Ref. 24 was removed and replaced with new references 24 and 25. The sentence was revised. Line 98- 99.
2. Lines 103-the end of the paragraph: The description misleads readers, especially those unfamiliar with this area. Revise totally. It has already been denied that the wild-type Is-PETase has better activity than PET hydrolases known up to 2016. Ref 30 did not try to improve Is-PETase. Even now, no mutants derived from Is-PETase surpassed thermophilic PET hydrolases. Is-PETase (Yoshida et al.) insisted that the enzyme is better at 30°C than thermophilic enzymes (unfair description) and even 30°C is higher than the temperature in marine environments. Ref. 31 is not based on the sequence of Is-PETase (based on the amino acid sequences based on various PET hydrolases, as the basic structure of Is-PETase is homologous to those of thermophilic enzymes). This part must be rewritten, for example, as follows: PET hydrolases are categorized into marine and terrestrial ones (Danso et al.). The major microorganisms in marine environment are --- and those in terrestrial environment are---. As PET hydrolases derived from mesophilic microorganisms, SM14est -----.	The sentence “Is-PETase appears to be ...” was removed. Ref 30 was removed. The fact that Is-PETase works efficiently under mesophilic conditions was described. Recently reported Cryptosporangium aurantiacum (CaPETase) with high thermostability was introduced here. Line 100- 109.
3. Line 142: Grammatically wrong with “detailed characterized”.	Revised accordingly in line 141- 143 with “detailed” deleted.
Results 1. Lines 253-254: distinct from MHETase and Abhydrolase_5, but homologous to abhydrolase_3 including IsPETase	Thank you for your suggestion, but we don’t agree with you. Is-PETase belongs to Abhydrolase_5 based on the phylogenetic tree constructed in Fig. 3b. Danso et al. also confirmed that Is-PETase belongs to Abhydrolase_5 (Danso et al. Applied and Environmental Microbiology, 2018).
2. Line 257: BHET, not monomeric BHET	Revised accordingly.
3. Line 261: “monomeric products” are strange. Why are they monomeric? Monomeric is used, for example, like that MHET is a monomeric unit for PET. No TPA and BHET polymers exists. Therefore, they cannot be monomers for	Revised accordingly. “monomeric” in line 172, 259, 348, 410, 415, 417, 464, 465 and 545 were all removed.

their polymers. Authors confuse monomeric materials and small molecules. Revise the paper totally because erroneous words are used throughout the paper.	
4. Lines 269 and 275: relatively high, not relative high. English edition by a professional native speaker (available) is requisite. There are too many grammatical errors.	Revised accordingly in line 258, 266, 273, 282, 298, 307 and 353.
5. Line 275: relative high enzymatic?	relatively high enzymatic activity
6. Lines 276-280 : ?	This sentence was revised to be concise. Line 274-276.
7. Lines 286-287: Probably with E. coli. Then, suggest this.	Yes, with purified PET esterase (OQN32_06240). We clarified this in this sentence. Line 280- 282.
8. PET plastic is strange. PET is enough. We never express PE, PVC and any plastics like PE plastic, PVC plastic etc.	We checked through the whole manuscript and replaced “PET plastic” with “PET” to make it more concise.
Discussion 1. This section is categorized in subsection, but I do not think that subsections are appropriate in Discussion. The description of the first subsection, especially lines 390-413, is not discussion but results (should be included in Results).	During the last review, some reviewer pointed out that this is a proposed degradation mechanism and not all of the claims such as trapping of PET monomers had been experimentally confirmed. As a result, this section was moved to the “Discussion” section.
2. Line 452: Is-PETase (WT) has no H MHET hydrolase activity.	Revised accordingly. Line 446.
3. Line 455: ---pH 8, which is not the optimum pH to activate MHET hydrolase---- Optimum pH cannot be used for activation.	Revised as follow: The MHET hydrolase activity of PET esterase from R. pyridinivorans P23 is generally inactive in seawater (about pH 8).
4. Line 462: The section title is inappropriate.	The title was revised as “PET esterase (OQN32_06240) is an important member of the PET hydrolase family”.
5. Line 464: As described above, PET esterase is close to Abhydrolase_3 including Is-PETase.	According to the phylogenetic tree, PET esterase (OQN32_06240) belongs to Abhydrolase_3 family while IS-PETase belongs to Abhydrolase_5 family. 6. Lines 472-475: Rewrite the sentence. Show the reference for Ple628 and PE-H, separately and respectively.	Revised accordingly. Line 467- 468.
7. Lines 491-495: Rewrite. How do you think to apply your strain for the bioremediation? The paper suggests the slow decay of PET in marine environments,	Although PET biodegradation by R. pyridinivorans P23 was relatively slow. However, there are large amount of similar PET degradation and assimilation functional

	microbial groups in the marine environment. They would contribute to the bioremediation of PET plastic pollution in the marine environment naturally.
Materials and Methods 1. Line 521: The 2216 marine broth is better, as figures at the top of a sentence must be written in alphabet like twenty-two hundreds and sixteen	Revised accordingly. Line 513.